# A pyroptosis-related gene signature for the diagnosis of acute pancreatitis

Yuting Wang[1], Jun Li[1], Zhongsu Yu[1], Shuyuan Li[1], Yuxia Chen[1], Yun Pan[1], Liangping Cheng[1,2]*, Guangyuan Yu[1,3]*

**1** Department of Gastroenterology, Children's Hospital of Chongqing Medical University, National Clinical Research Center for Children and Adolescents' Health and Diseases, Ministry of Education Key Laboratory of Child Development and Disorders, Chongqing, China, **2** National Clinical Key Cardiovascular Specialty; Key Laboratory of Children's Important Organ Development and Diseases of Chongqing Municipal Health Commission, Chongqing, China, **3** Chongqing Key Laboratory of Pediatric Metabolism and Inflammatory Diseases, Chongqing, China

☯ Yuting Wang and Jun Li are co-first authors of this paper.
* 486312@hospital.cqmu.edu.cn (LC); guangyuanyu90@126.com (GY)

## Abstract

Acute pancreatitis (AP) is a severe inflammatory disorder in which pyroptosis—a pro-inflammatory form of programmed cell death—may contribute to pathogenesis. However, the complete transcriptional profile of pyroptosis-related genes (PRGs) in AP and their potential as diagnostic biomarkers remain underexplored. This study aimed to systematically characterize pyroptosis-associated transcriptional signatures and identify the reliable biomarkers for diagnostic purposes. Three transcriptomic datasets from murine AP models were integrated to identify pyroptosis-related differentially expressed genes (PRDEGs). Functional enrichment and immune cell infiltration analyses were conducted to elucidate the biological pathways and immune microenvironment alterations associated with these genes. mRNA-transcription factor (TF) and mRNA-microRNA (miRNA) regulatory networks were constructed to investigate underlying molecular interactions. Machine learning techniques, including support vector machine (SVM) and least absolute shrinkage and selection operator (LASSO), were applied for feature selection, leading to the identification of key diagnostic markers and the development of a logistic regression model. The regression model were then assessed using an independent cohort of human peripheral blood samples. Eleven PRDEGs were identified, with enrichment observed in processes such as cytoskeletal organization, cell-substrate adhesion, and critical inflammatory signaling pathways, including MAPK and NF-κB. Immune infiltration analysis revealed significant correlations between these PRDEGs and various immune cell subsets, particularly M1 macrophages, Treg cells, and monocytes. A four-gene diagnostic signature, comprising ANXA3, IQGAP1, RELA, and VTN, was established through SVM and LASSO analysis. In the independent human cohort, the fixed-coefficient four-gene model demonstrated reduced discrimination, which likely reflects interspecies and tissue-specific variations. However, after

**Data availability statement:** All relevant data are publicly available without restriction. Raw microarray and RNA-seq datasets were obtained from the NCBI Gene Expression Omnibus (GEO) repository under accession numbers GSE109227, GSE65146, GSE121038 and GSE194331. Other supporting data and analysis details are provided in the Supporting Information files accompanying the paper.

**Funding:** This work was supported by the National Natural Science Foundation of China (https://www.nsfc.gov.cn, Grant No. 82200011 to YGY). The funder had no role in study design, data collection and analysis, decision to publish, or preparation of the manuscript.

**Competing interests:** The authors have no conflicts to disclose.

optimizing the model to exclude non-significant predictors, a refined two-gene signature (ANXA3 and IQGAP1) exhibited improved accuracy, with excellent calibration and clinical net benefit. This study offers a comprehensive transcriptomic analysis of the pyroptosis-mediated landscape and immune microenvironment in AP. An optimized two-gene signature, comprising ANXA3 and IQGAP1, was validated in a human cohort with superior accuracy, reflecting critical disruptions in inflammatory pathways and cytoskeletal organization. Notably, ANXA3 demonstrated potential for stratifying disease severity. Although these markers hold potential for molecular diagnosis, further prospective studies are essential to establish their clinical specificity and generalizability across diverse populations.

## Introduction

AP is a sudden inflammation of the pancreas that can vary in severity from mild, self-limiting conditions to severe, life-threatening illnesses. It is a prevalent gastrointestinal cause of hospital admissions globally [1], and the mortality rate for severe acute pancreatitis can reach as high as 30% [2]. Despite advancements in medical care, the management of AP primarily consists of supportive measures, as there are currently no specific pharmacological treatments available [3]. This underscores the pressing need for innovative diagnostic markers and therapeutic targets to enhance the clinical outcomes of patients with AP.

Pyroptosis has hallmarks of cell swelling, lysis, and secretion of pro-inflammatory cytokines, contributing to the inflammatory response [4]. Recent studies have highlighted the significant role of pyroptosis in various inflammatory diseases such as sepsis, atherosclerosis, and neurodegenerative disorders [5–7]. Despite this, the involvement of pyroptosis in acute pancreatitis remains poorly understood. Exploring the mechanisms of pyroptosis in acute pancreatitis could may offer new insights into the disease's pathophysiology and potentially identify novel therapeutic targets.

The objective of this study was to explore the involvement of pyroptosis in acute pancreatitis through the identification of PRDEGs. Utilizing bioinformatics methodologies, we conducted comprehensive analyses of gene expression datasets obtained from the Gene Expression Omnibus (GEO) database. Subsequently, we performed functional enrichment analyses and developed a four-gene diagnostic model using machine-learning approaches, which was then externally validated in an independent human RNA-seq cohort.

## Methods

### Data acquisition

The GEOquery package [8] from the R programming language was used to download data from the GEO database. Three datasets—GSE109227 [9], GSE65146, and GSE121038—focused on AP were obtained. These datasets, derived from Mus musculus, include pancreas tissue samples. A detailed overview of these datasets is provided in Table 1.

**Table 1. GEO microarray chip information.**

| | GSE109227 | GSE65146 | GSE121038 |
|---|---|---|---|
| Platform | GPL6246 | GPL6246 | GPL10787 |
| Species | Mus musculus | Mus musculus | Mus musculus |
| Tissue | Pancreas | Pancreas | Pancreas |
| Samples in AP group | 6 | 39 | 8 |
| Samples in Control group | 5 | 5 | 7 |
| Reference | PMID: 30139658 | \ | \ |

In the GSE109227 dataset, the GPL6246 chip platform was used, which included 6 AP samples and 5 control samples. Similarly, the GSE65146 dataset utilized the GPL6246 chip platform and contained 39 AP samples, 5 control samples, and 27 mutant mouse samples. The GSE121038 dataset, on the other hand, employed the GPL10787 chip platform and consisted of 8 AP samples and 7 control samples. For the purpose of this study, only the AP and control groups were considered.

We utilized the GeneCards database [10] (https://www.genecards.org/) to identify PRGs. By searching for the terms "Pyroptosis" and "Protein Coding," we selected genes with a relevance score > 1, identifying 405 PRGs. A subsequent search for "Pyroptosis" in the PubMed database (https://pubmed.ncbi.nlm.nih.gov/) retrieved 33 PRGs from the literature [11]. After removing duplicates, we combined the results, yielding a total of 413 human PRGs. These were cross-referenced with 334 mouse PRGs using the R package Homologene (Version 1.4.68.19.3.27).

The sva R package [12] (v3.50.0) was used to remove batch effects across the GSE109227, GSE65146, and GSE121038 datasets, generating a combined dataset comprising 53 AP cases and 17 controls. The integrated dataset was then normalized using the limma R package [13]. Probe annotations were harmonized and standardized across platforms. Finally, principal component analysis (PCA) was performed on the expression matrix to verify the effectiveness of batch-effect correction.

## Differentially expressed genes related to pyroptosis associated with AP

Based on the sample annotations in the GEO database, samples were assigned to either the control or AP group. Differential expression analysis between groups was performed in R using appropriate Bioconductor packages. Differentially Expressed Gene (DEGs) were defined as those with an absolute log2 fold change (|log2FC|) > 1 and adjusted P value < 0.05. Genes with log2FC > 1 were considered upregulated, whereas those with log2FC < −1 were considered downregulated. DEGs were visualized using volcano plots.

To identify PRDEGs associated with AP, PRGs were intersected with the DEGs identified from combined GEO datasets, with |log2FC| > 1 and adjusted P value < 0.05. A Venn diagram was then constructed to visualize the overlapping genes. The PRDEGs were further represented using a heatmap.

## GO and KEGG enrichment analysis

Gene Ontology (GO) analysis [14] is a widely used method for investigating functional enrichment across categories such as Biological Process (BP), Cellular Component (CC), and Molecular Function (MF). The Kyoto Encyclopedia of Genes and Genomes (KEGG) [15] is a well-established resource for understanding genomic data, biological pathways, diseases, and drugs. Enrichment analyses of PRGs were performed using the clusterProfiler package [16]. The criteria for significant differences were set at a false discovery rate (FDR, q-value) < 0.25 and adjusted P value < 0.05.

## Establishment of diagnostic model for AP

To develop diagnostic models for AP using the combined GEO datasets, we performed logistic regression to identify PRDEGs that discriminate AP from control samples. This analysis evaluated the association between the candidate predictors (PRDEGs) and the binary outcome (AP vs control). PRDEGs with $P < 0.05$ were considered statistically significant and were included in the final logistic regression model; the resulting effect estimates were visualized using a forest plot.

Utilizing PRDEGs identified through logistic regression, SVM [17] was employed for establishing an optimized predictive model. The selection of the number of genes that provided the highest accuracy and lowest error rate was a key aspect of model optimization.

Subsequently, the study applied the R package glmnet [18] (Version 4.1–8) to perform LASSO regression analysis. The seed was set to 600, and family = "binomial" was used as the parameter. LASSO regression analysis included a term known as penalty (lambda × absolute coefficient of slope) in the calculation of linear regression, which helped to reduce overfitting and improve the model's capacity for applicability. The results were visualized using trajectory plots and model diagnostic plots. Ultimately, an AP diagnostic model was constructed by integrating the PRDEGs selected by LASSO.

RiskScore was obtained with risk coefficients from the LASSO regression analysis using the following formula: [].

**Validation of the diagnostic model for AP.** ROC curves and AUC values were generated using the pROC package [19] to assess the diagnostic performance of the AP model RiskScore. Subsequently, a nomogram was constructed based on the logistic regression model to visualize the contribution and combined effects of the variables on a two-dimensional coordinate scale [20].

The correctness and identification capabilities of an AP diagnostic model created using LASSO regression analysis will be assessed using calibration analysis. A Calibration Curve was established to visualize the consistency of expected likelihoods and actual results.

Furthermore, DCA [21] made by R package ggDCA (Version 1.1) was performed to produce DCA maps based on the model genes in the combined GEO Datasets. To evaluate the overall clinical utility of the diagnostic model in healthcare decision-making, DCA was performed.

## Functional similarity (Friends) analysis

The quantitative assessment of gene and genome similarity through semantic comparison of GO annotations has become a foundational approach in bioinformatics analyses. To assess the functional similarity of the model genes, referred to as Friends analysis, the R package GOSemSim [22] was employed.

## DEG verification and ROC curve

To analyze the differential expression of PRDEGs between control and AP samples, a group comparison map was constructed, enabling a visual evaluation of the DEGs' expression levels. ROC curves for the PRDEGs were then plotted using the R package pROC, and AUC values were calculated. AP samples selected from the GEO Database were classified into HighRisk and LowRisk groups using the median RiskScore from the AP diagnostic models. To further evaluate the DEGs of model genes, a group comparison map was drawn to illustrate gene expression differences between the two groups. ROC curves for the model genes were plotted using the pROC package, and AUC values were calculated to assess their diagnostic value in AP progression.

## Gene set enrichment analysis (GSEA)

GSEA [23] was performed to evaluate gene distribution trends within predefined gene sets, ranked by their correlation with the phenotype, to determine their contribution to AP. In this study, genes from the Combined GEO Datasets were ranked by their logFC values. GSEA was then performed using the R package clusterProfiler, with the following parameters:

seed = 2020, number of computations = 1000, minimum number of genes in each gene set = 10, and maximum number of genes in each gene set = 500.

The gene sets for GSEA were from MSigDB [24](https://www.gsea-msigdb.org/gsea/msigdb) with screening criteria of adj.p < 0.05 and q < 0.25.

AP samples from the Combined GEO Datasets were categorized into HighRisk and LowRisk groups based on the median RiskScore of the diagnostic model. DEGs were analyzed using the limma R package, and volcano plots were generated to display genes with |logFC| > 1 and adj.p < 0.05 in the HighRisk and LowRisk groups. The R package pheatmap was employed to create heatmaps of these gene expression values.

Finally, GSEA was performed on all genes in the AP samples using clusterProfiler, with gene sets from the MSigDB c2.cp.all.v2022.1.Hs.symbols.gmt. The screening criteria for GSEA were adj.p < 0.05 and q < 0.25, with the minimum and maximum numbers of genes in each gene set set to 10 and 500, respectively.

## Protein-Protein Interaction (PPI) Network

Through association with model genes in the post-transcriptional phase, TFs regulate gene expression. The ChIPBase database [25] (http://rna.sysu.edu.cn/chipbase/) was employed to identify TFs and examine their regulatory functions on the model genes. Cytoscape [21] was used to visualize the mRNA-TF regulatory network.

MiRNAs play a critical ecological role in cellular growth and progression by controlling multiple target genes and regulating the same gene through several miRNAs. MiRNAs associated with the model genes were obtained from the StarBase v3.0 database [26], allowing for the investigation of miRNA-gene associations. Cytoscape software was subsequently used to illustrate the mRNA-miRNA regulatory network.

## Immune infiltration analysis

CIBERSORT [27] employs linear support vector regression to deconvolute transcriptomic expression matrices and estimate the composition and abundance of immune cells in mixed cell populations. By applying the CIBERSORT algorithm combined with immune cell gene matrix features and filtering for data with an immune cell enrichment score greater than zero, the immune cell infiltration results in the Combined Datasets were obtained. The proportion of immune cells was visualized using a bar chart.

Group comparison plots were generated using the R package ggplot2 to visualize differences in immune cell composition between control and AP samples. Spearman correlation analysis was performed to assess relationships, and the results were visualized in a correlation heatmap. Spearman correlation was also applied to evaluate the association between immune cells and model genes, which was then visualized using a bubble plot.

Using the same procedures described above, immune cell infiltration was further compared between the LowRisk and HighRisk AP subgroups.

## Validation of the diagnostic model for AP in a human RNA-seq cohort (GSE194331)

We downloaded the raw count matrix for the independent human RNA-seq cohort GSE194331, consisting of adult patients (AP = 87, Control = 32), from the GEO database. The counts were normalized using DESeq2 size factors, and the normalized expression values were log2-transformed for downstream analyses. The samples were grouped into the AP and control cohorts based on the phenotype annotations in the GEO metadata. To visualize predefined genes of interest, faceted boxplots were generated for the 11 PRDEGs, and group differences (AP vs Control) were assessed using the Mann-Whitney U test.

To assess the cross-domain transportability, the four-gene diagnostic model, previously derived from the murine discovery cohort, was applied to the human blood cohort using fixed coefficients (without re-estimation). Diagnostic performance was evaluated using ROC analysis and AUC. Due to the reduced discrimination observed with fixed coefficients, an exploratory optimization of the model was performed. The Akaike Information Criterion (AIC) was first used to compare the

fit and parsimony between the full four-gene model and the reduced two-gene model (ANXA3, IQGAP1). Subsequently, the coefficients for the optimal gene signature were re-estimated using logistic regression. Diagnostic discrimination and clinical utility were assessed via ROC analysis and DCA, respectively, following the procedures described in the 'Validation of the diagnostic model for AP' section. Calibration was quantified by the Brier score and calibration slope, alongside visual inspection of calibration plots (Bootstrap B = 1000) and the Hosmer-Lemeshow test. To prevent overfitting and ensure the reliability of the model recalibration, a rigorous split-sample validation strategy was implemented. The external human cohort (N = 119) was randomly partitioned into a training set (50%, n = 60) and an independent testing set (50%, n = 59). The coefficients for the optimal two-gene signature were re-estimated using logistic regression exclusively within the training set. Diagnostic discrimination was then verified on the unseen testing set using ROC analysis and AUC to confirm that the high diagnostic accuracy was not an artifact of overfitting.

For severity-related analyses within AP, samples were stratified into Mild AP, Moderate AP, and Severe AP based on phenotype descriptors. The expression of the four model genes across severity categories was visualized using faceted boxplots, with overall differences assessed by using Mann-Whitney U test. ANXA3 was additionally evaluated as a single-gene severity marker using ROC analysis.

### Statistical analysis

All analyses were conducted using R (v4.3.0). Continuous variables were summarized as mean ± SD or median (IQR), based on the Shapiro–Wilk normality test, and were compared using the independent Student's t-test (after testing for homoscedasticity) or the Mann–Whitney U test as appropriate. For comparisons across more than two groups, one-way ANOVA or the Kruskal–Wallis test with Dunn's post-hoc comparisons was used. Gene–gene associations were evaluated using Spearman's rank correlation. For prespecified candidate genes, p values were adjusted for multiple testing using the Benjamini–Hochberg (BH) method and reported as adjusted p (adj.p), with significance set at adj.p < 0.05 (or two-sided p < 0.05). And all R scripts have been made publicly available via our GitHub repository: https://github.com/guangyuanyu90-ai/AP-diagnostic-model.

### Ethics statement

All data analyzed in this study were obtained from publicly available, de-identified datasets in the GEO database. Because this study involved only the secondary analysis of anonymous public data and did not include direct human participant involvement or animal experiments, additional institutional ethical approval and informed consent were not required.

## Results

### Technology roadmap

Fig 1 outlines the comprehensive pipeline used to identify AP-related pyroptosis signatures and develop a diagnostic model. Three discovery datasets were merged and normalized to form the training cohort. DEGs and PRGs were integrated to identify PRDEGs, followed by GO and KEGG enrichment analyses. A four-gene diagnostic model was derived and subsequently used for downstream analyses in the discovery cohort, including risk grouping, immune infiltration by CIBERSORT, regulatory network construction, and GSEA. An independent human RNA-seq cohort was utilized for external validation. The expression levels of PRDEGs were checked, and the four-gene model was re-evaluated through ROC, Calibration and DCA.

### Merging of AP datasets

Batch effects in the GSE109227, GSE65146, and GSE121038 datasets were eliminated using the "sva" R package, resulting in combined GEO datasets. A distribution boxplot was generated to analyze dataset levels, and a PCA plot was created to examine data distribution. Both the boxplot and PCA plot demonstrated effective batch effect removal in the AP dataset samples (Fig 2).

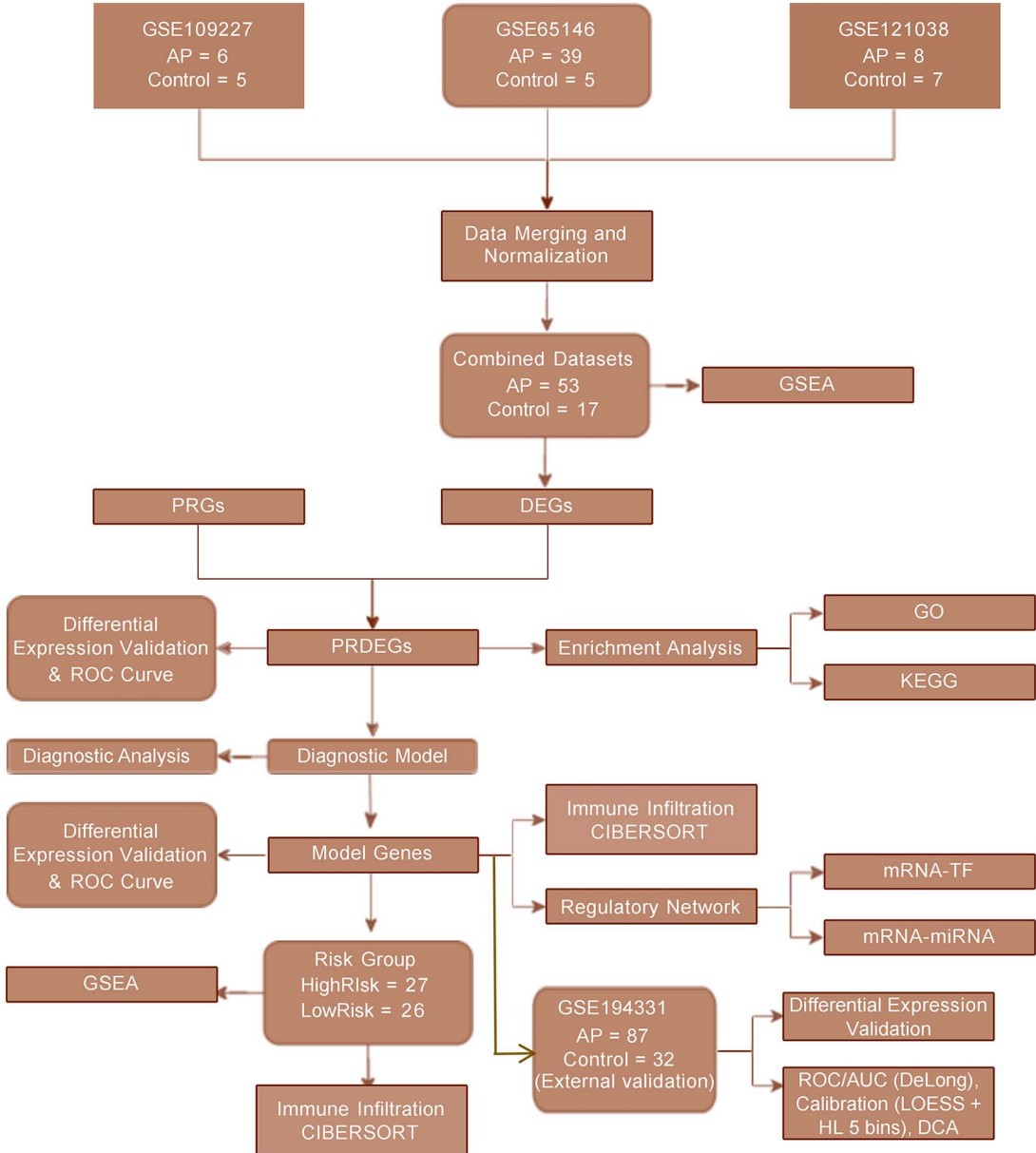

**Fig 1. Flowchart of the study design and analytical workflow.** AP, acute pancreatitis; PRGs, pyroptosis-related genes; DEGs, differentially expressed genes; PRDEGs, pyroptosis-related differentially expressed genes; GO, Gene Ontology; KEGG, Kyoto Encyclopedia of Genes and Genomes; GSEA, gene set enrichment analysis; CIBERSORT, Cell-type Identification By Estimating Relative Subsets Of RNA Transcripts; ROC/AUC, receiver operating characteristic/area under the curve; DCA, decision curve analysis; TF, transcription factor.

## Pyroptosis associated DEGs in AP

The combined GEO datasets, including AP and control groups, were analyzed for gene level differences. Differential analysis identified 213 DEGs, meeting the criteria of |logFC| > 1 and adj.p < 0.05. Among these, 177 genes were up-regulated (logFC > 1 and adj.p < 0.05) and 36 were down-regulated. A volcano plot visualized these findings.

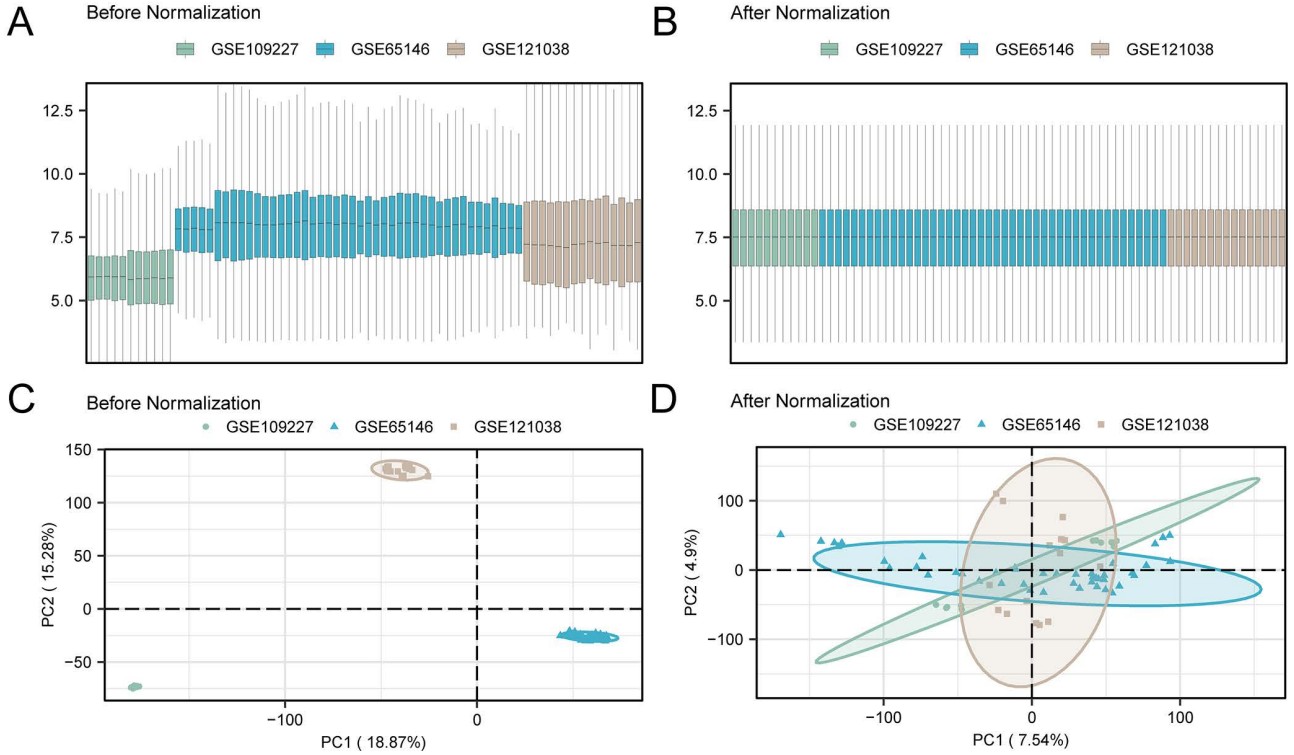

**Fig 2. Batch effects removal of GSE109227, GSE65146 and GSE121038.** (A) Boxplot before batch effect correction. (B) Boxplot after batch effect correction.(C) PCA plot before batch effect correction. (D) PCA plot after batch effect correction.GSE109227 is shown in green, GSE65146 in blue, and GSE121038 in brown.

To identify PRDEGs, DEGs with |logFC|>1 and adj.p<0.05 were intersected with PRGs. This intersection yielded 11 PRDEGs: Rela, Iqgap1, Actn4, Flna, Anxa3, Vtn, Lcn2, Mpeg1, Pah, Car9, and Pyhin1. A Venn diagram was used to illustrate the overlap. The expression levels of these PRDEGs were assessed, and heat maps are presented in Fig 3C.

### Verification of PRDEGs and ROC curve

Variable PRDEGs were evaluated in the combined GEO datasets, with a group comparison displaying the differential expression of PRDEGs between AP and control samples. The analysis revealed statistically significant expression levels of the 11 PRDEGs (p<0.01), which included Rela, Iqgap1, Actn4, Flna, Anxa3, Vtn, Lcn2, Mpeg1, Pah, Car9, and Pyhin1 (Fig 4).

ROC curve analysis was performed for PRDEGs, with results showing high diagnostic accuracy (AUC>0.9) for five PRDEGs—Rela, Iqgap1, Actn4, Flna, and Anxa3. Vtn, Lcn2, Mpeg1, Pah, Car9, and Pyhin1 demonstrated moderate accuracy (AUC 0.7–0.9) in distinguishing AP from control samples.

### Enrichment analysis

GO and KEGG enrichment analyses were conducted to investigate the relationships between the 11 PRDEGs and AP. As shown in Table 2, the 11 PRDEGs were primarily enriched in the following categories:

1. BP: Regulation of cell-substrate adhesion; Cell-substrate adhesion; Ameboidal-type cell migration; Positive regulation of cell-substrate adhesion; Protein localization to cell-cell junction.

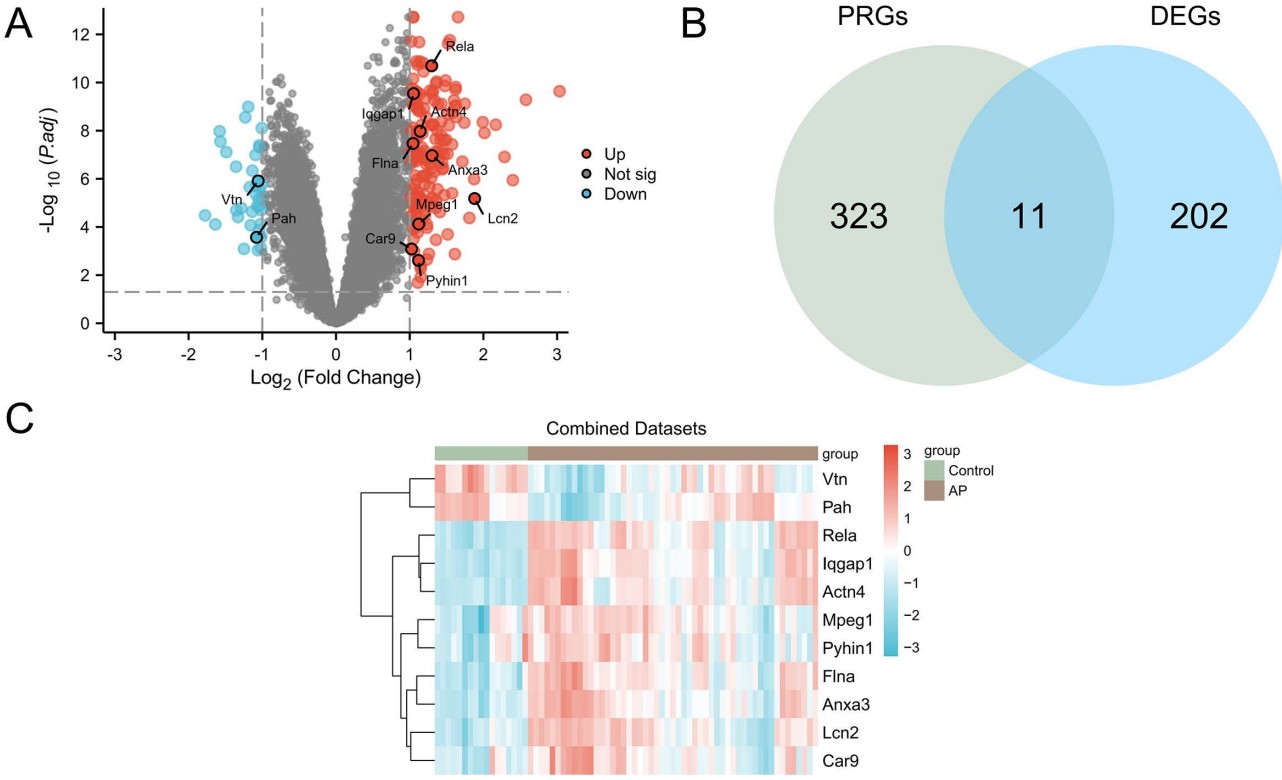

**Fig 3. DGE analysis.** (A)Volcano plot of DEGs between the AP and control groups. (B) Venn diagram showing the overlap between DEGs and PRGs. (C) Heatmap of the 11 PRDEGs. Brown indicates the AP group and green indicates the control group; red indicates relatively high expression and blue indicates relatively low expression.

2. CC: Cortical cytoskeleton; Cell cortex; Cortical actin cytoskeleton; Actin filament bundle; Actin cytoskeleton.

3. MF: Actin filament binding; Actin binding; Chromatin DNA binding; Transmembrane receptor protein tyrosine kinase activity; Iron ion binding.

4. KEGG: Focal adhesion (Mus musculus); Proteoglycans in cancer (Mus musculus)

The results of the enrichment analyses were visualized through histograms (Fig 5) and network diagrams for BP, CC, MF, and KEGG. Larger nodes represented terms containing additional molecules, while lines depicted relevant molecular interactions. These analyses highlighted the significant involvement of genes in regulating cell-substrate adhesion and ameboid-type cell migration.

**GSEA for AP**

To evaluate the impact of the identified genes on AP in the Combined GEO Datasets, GSEA was conducted to examine the levels of all genes and the BP, CC, and MF involved. The results are summarized in Table 3 and visualized in Fig 6A.

GSEA revealed significant enrichment of genes in several key pathways and functions, which including: MAPK Pathway, NF-κB Pathway, TGF-β Pathway, PI3K-Akt Signaling Pathway, providing novel insights into the molecular mechanisms underlying AP and highlighting the BP, CC, and MF influenced by gene expression changes in AP.

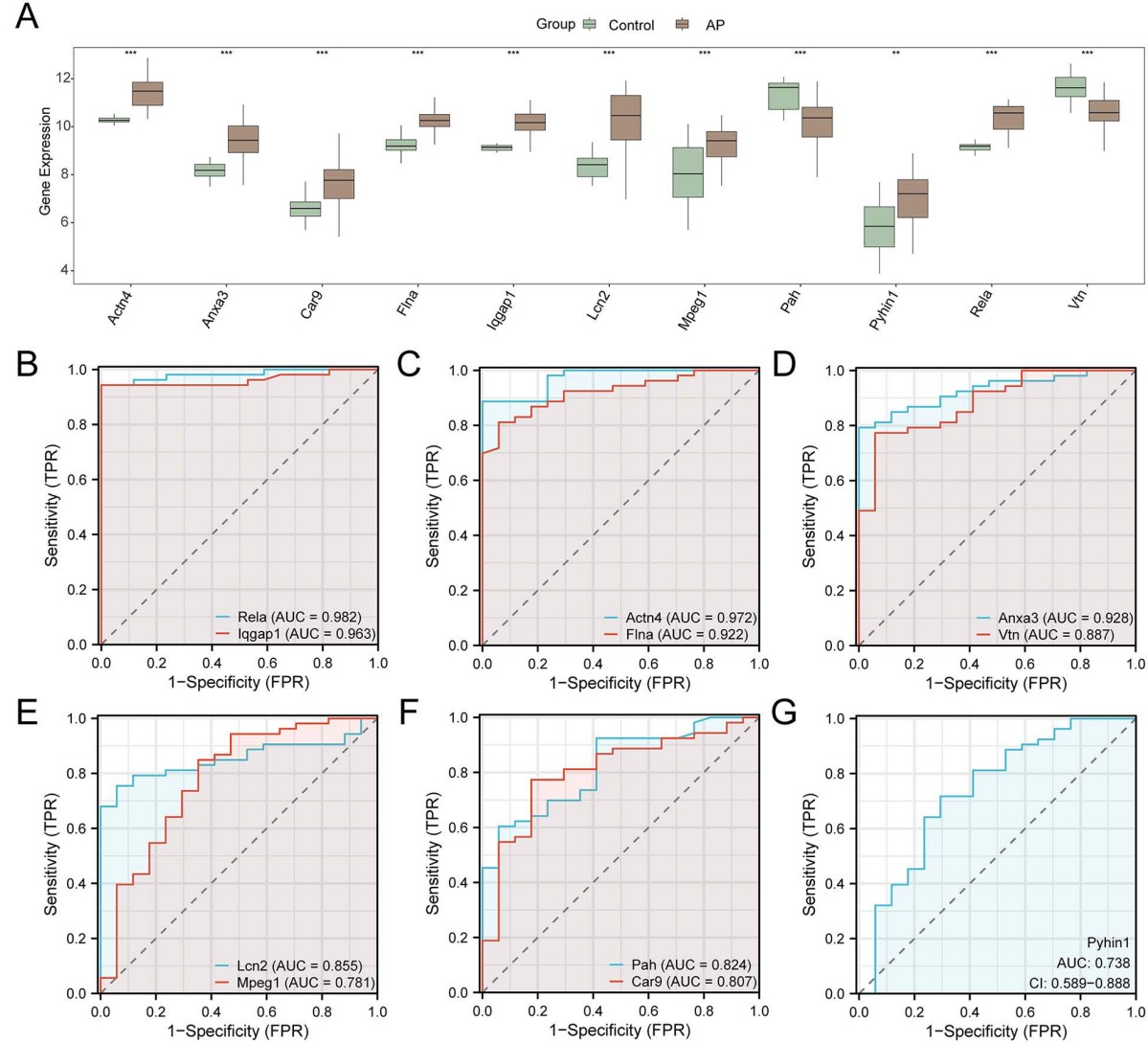

**Fig 4. Differential expression validation and ROC analysis.** (A) (A) Gene Expression by Group. Boxplots show the expression levels of 11 genes (Actn4, Anxa3, Car9, Flna, Iqgap1, Lcn2, Mpeg1, Pah, Pyhin1, Rela, and Vtn) in Control and AP groups. Statistical significance was assessed using the Kruskal-Wallis test, with p-values indicated by asterisks (***p < 0.001, **p < 0.01). (B-G) ROC Curves for Gene Markers. ROC curves show the diagnostic performance of individual genes in distinguishing AP from Control groups. The area AUC is reported for each gene, with 95% confidence intervals indicated where applicable.

### Establishment of a diagnostic model for AP

To assess the diagnostic potential of the 11 PRDEGs in AP, logistic regression analysis was performed and visualized using a Forest Plot. All 11 PRDEGs were significantly associated with AP (p < 0.05), including Rela, Iqgap1, Actn4, Flna, Anxa3, Vtn, Lcn2, Mpeg1, Pah, Car9, and Pyhin1.

An SVM model was then constructed using the 11 PRDEGs to determine the number of genes that maximized accuracy and minimized error. The analysis revealed that the highest accuracy occurred with the inclusion of ten genes in the SVM model.

 

**Table 2. Enrichment analysis for PRDEGs.**

| ONTOLOGY | ID | Description | GeneRatio | BgRatio | pvalue | p.adjust | qvalue |
|---|---|---|---|---|---|---|---|
| BP | GO:0010810 | regulation of cell-substrate adhesion | 4/10 | 230/28564 | 8.28 e-07 | 4.99 e-04 | 1.86 e-04 |
| BP | GO:0031589 | cell-substrate adhesion | 4/10 | 369/28564 | 5.41 e-06 | 1.63 e-03 | 6.07 e-04 |
| BP | GO:0001667 | ameboidal-type cell migration | 4/10 | 470/28564 | 1.40 e-05 | 2.12 e-03 | 7.89 e-04 |
| BP | GO:0010811 | positive regulation of cell-substrate adhesion | 3/10 | 142/28564 | 1.41 e-05 | 2.12 e-03 | 7.89 e-04 |
| BP | GO:0150105 | protein localization to cell-cell junction | 2/10 | 24/28564 | 3.03 e-05 | 3.66 e-03 | 1.36 e-03 |
| CC | GO:0030863 | cortical cytoskeleton | 3/9 | 119/28585 | 5.80 e-06 | 3.60 e-04 | 1.10 e-04 |
| CC | GO:0005938 | cell cortex | 3/9 | 300/28585 | 9.17 e-05 | 2.84 e-03 | 8.69 e-04 |
| CC | GO:0030864 | cortical actin cytoskeleton | 2/9 | 84/28585 | 3.03 e-04 | 5.12 e-03 | 1.57 e-03 |
| CC | GO:0032432 | actin filament bundle | 2/9 | 97/28585 | 4.04 e-04 | 5.12 e-03 | 1.57 e-03 |
| CC | GO:0015629 | actin cytoskeleton | 3/9 | 500/28585 | 4.13 e-04 | 5.12 e-03 | 1.57 e-03 |
| MF | GO:0051015 | actin filament binding | 3/9 | 221/28171 | 3.86 e-05 | 3.25 e-03 | 1.26 e-03 |
| MF | GO:0003779 | actin binding | 3/9 | 447/28171 | 3.10 e-04 | 1.30 e-02 | 5.07 e-03 |
| MF | GO:0031490 | chromatin DNA binding | 2/9 | 115/28171 | 5.84 e-04 | 1.63 e-02 | 6.35 e-03 |
| MF | GO:0044325 | transmembrane transporter binding | 2/9 | 152/28171 | 1.02 e-03 | 2.13 e-02 | 8.27 e-03 |
| MF | GO:0005506 | iron ion binding | 2/9 | 170/28171 | 1.27 e-03 | 2.13 e-02 | 8.27 e-03 |
| KEGG | mmu04510 | Focal adhesion – Mus musculus (house mouse) | 3/8 | 202/9710 | 4.60 e-04 | 2.01 e-02 | 1.67 e-02 |
| KEGG | mmu05205 | Proteoglycans in cancer – Mus musculus (house mouse) | 3/8 | 204/9710 | 4.73 e-04 | 2.01 e-02 | 1.67 e-02 |

Subsequently, an AP diagnostic model was developed using LASSO regression analysis incorporating the 11 PRDEGs identified through SVM. Visualizations of the LASSO regression model (Fig 7D) and the LASSO variable trajectory (Fig 7E) were generated. The model identified four key PRDEGs—Vtn, Anxa3, Rela, and Iqgap1—as the most relevant genes for inclusion in the diagnostic model.

The RiskScore for the LASSO model was calculated based on the risk coefficients from the LASSO regression analysis using the following formula: RiskScore = Vtn × (−0.175) + Anxa3 × 0.21 + Rela × 2.754 + Iqgap1 × 0.319.

## Validation and Friends analysis of the AP diagnostic model

ROC curves were plotted using the RiskScore with the R package pROC. The ROC curves (Fig 8A) indicated that the RiskScore had high diagnostic accuracy, with an AUC > 0.9.

Nomograms were generated to visualize the interrelationships of the model genes in the diagnostic model (Fig 8B). The analysis showed that Rela had the most significant contribution to the AP diagnostic model, while Anxa3 showed a comparatively lower effect.

Calibration analysis was conducted to assess the model's precision and specificity. The calibration curve (Fig 8C) demonstrated that the observed and expected probabilities were closely aligned, indicating robust predictive power.

DCA was performed to evaluate the clinical utility of the model (Fig 8D). The DCA results showed that the model's net benefit remained higher than the "All Positive" and "All Negative" lines, indicating excellent clinical performance.

Functional similarity (Friends) analysis scores were used to identify key genes involved in the biological processes of AP. Fig 8E highlighted Iqgap1 as a pivotal gene in AP, with a score closest to the critical value (cut-off = 0.55).

## Verification and ROC curve of model genes

To examine the levels of the model genes in AP samples, a group comparison (Fig 9A) was performed, showing significant differential expression of the four model genes (Vtn, Anxa3, Rela, and Iqgap1) between the HighRisk and LowRisk

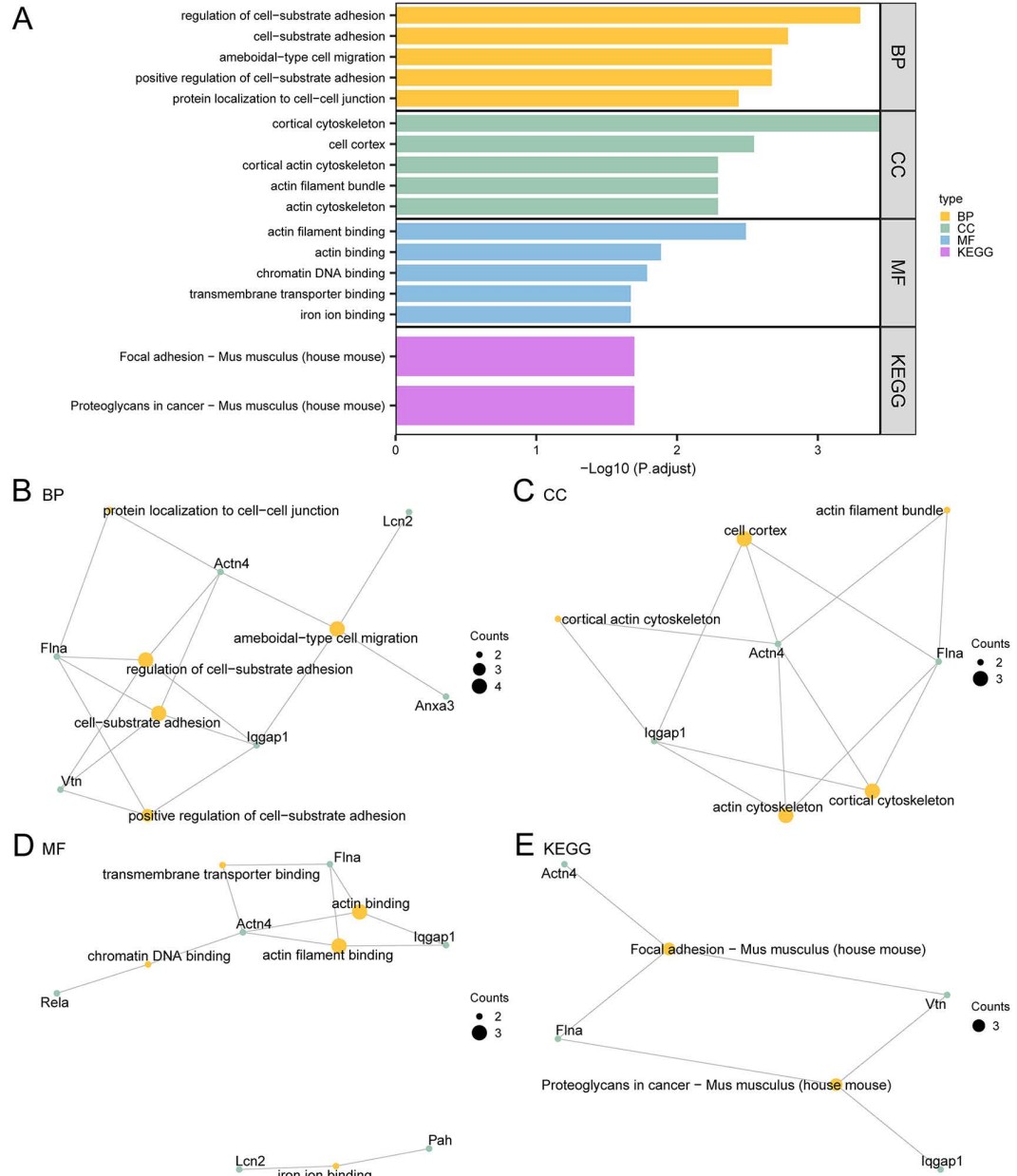

**Fig 5. GO and KEGG enrichment analysis for PRDEGs.** (A) Bar chart showing the enriched GO terms and KEGG pathways of PRDEGs, including BP, CC, MF, and KEGG categories. (B–E) Network plots of enriched BP (B), CC (C), MF (D), and KEGG pathways (E). Orange nodes represent enriched terms or pathways, and green nodes represent genes; connecting lines indicate gene-term associations. Significance thresholds were set at adj.p<0.05 and q<0.25.

groups. The expression levels of these genes were significantly different (p<0.001), further supporting their relevance in the diagnostic model.

ROC curves (Fig 9B-9E) demonstrated that two model genes, Iqgap1 and Rela, exhibited high diagnostic accuracy (AUC>0.9). In contrast, the expression levels of Anxa3 and Vtn showed moderate accuracy (0.7<AUC<0.9) in distinguishing between High Risk and Low Risk groups.

**Table 3. GSEA for combined datasets.**

| ID | Set Size | Enrichment Score | NES | p value | p.adjust | q value |
|---|---|---|---|---|---|---|
| REACTOME_TOLL_LIKE_RECEPTOR_CASCADES | 138 | 6.29 e-01 | 2.54 e+00 | 1.32 e-03 | 9.73 e-03 | 5.49 e-03 |
| REACTOME_INTERFERON_SIGNALING | 146 | 6.17 e-01 | 2.52 e+00 | 1.32 e-03 | 9.73 e-03 | 5.49 e-03 |
| KEGG_PATHOGENIC_ESCHERICHIA_COLI_INFECTION | 41 | 7.61 e-01 | 2.51 e+00 | 1.57 e-03 | 9.73 e-03 | 5.49 e-03 |
| WP_PATHOGENIC_ESCHERICHIA_COLI_INFECTION | 41 | 7.61 e-01 | 2.51 e+00 | 1.57 e-03 | 9.73 e-03 | 5.49 e-03 |
| REACTOME_MYD88_INDEPENDENT_TLR4_CASCADE | 93 | 6.56 e-01 | 2.50 e+00 | 1.40 e-03 | 9.73 e-03 | 5.49 e-03 |
| PID_PDGFRB_PATHWAY | 115 | 6.35 e-01 | 2.50 e+00 | 1.37 e-03 | 9.73 e-03 | 5.49 e-03 |
| WP_VEGFAVEGFR2_SIGNALING_PATHWAY | 373 | 5.50 e-01 | 2.48 e+00 | 1.16 e-03 | 9.73 e-03 | 5.49 e-03 |
| REACTOME_RHO_GTPASES_ACTIVATE_FORMINS | 104 | 6.37 e-01 | 2.46 e+00 | 1.39 e-03 | 9.73 e-03 | 5.49 e-03 |
| WP_TOLLLIKE_RECEPTOR_SIGNALING_PATHWAY | 88 | 6.42 e-01 | 2.43 e+00 | 1.39 e-03 | 9.73 e-03 | 5.49 e-03 |
| REACTOME_TOLL_LIKE_RECEPTOR_9_TLR9_CASCADE | 92 | 6.36 e-01 | 2.43 e+00 | 1.38 e-03 | 9.73 e-03 | 5.49 e-03 |
| REACTOME_TOLL_LIKE_RECEPTOR_TLR1_TLR2_CASCADE | 102 | 6.30 e-01 | 2.43 e+00 | 1.39 e-03 | 9.73 e-03 | 5.49 e-03 |
| WP_EBOLA_VIRUS_INFECTION_IN_HOST | 113 | 6.19 e-01 | 2.43 e+00 | 1.38 e-03 | 9.73 e-03 | 5.49 e-03 |
| WP_INTEGRINMEDIATED_CELL_ADHESION | 87 | 6.43 e-01 | 2.42 e+00 | 1.40 e-03 | 9.73 e-03 | 5.49 e-03 |
| KEGG_TOLL_LIKE_RECEPTOR_SIGNALING_PATHWAY | 87 | 6.41 e-01 | 2.41 e+00 | 1.40 e-03 | 9.73 e-03 | 5.49 e-03 |
| KEGG_LEISHMANIA_INFECTION | 59 | 6.81 e-01 | 2.41 e+00 | 1.47 e-03 | 9.73 e-03 | 5.49 e-03 |
| REACTOME_SIGNALING_BY_INTERLEUKINS | 373 | 5.35 e-01 | 2.41 e+00 | 1.16 e-03 | 9.73 e-03 | 5.49 e-03 |
| BIOCARTA_MAPK_PATHWAY | 78 | 5.97 e-01 | 2.20 e+00 | 1.43 e-03 | 9.73 e-03 | 5.49 e-03 |
| BIOCARTA_NFKB_PATHWAY | 19 | 7.25 e-01 | 2.02 e+00 | 1.69 e-03 | 9.73 e-03 | 5.49 e-03 |
| BIOCARTA_TGFB_PATHWAY | 19 | 7.13 e-01 | 1.99 e+00 | 1.69 e-03 | 9.73 e-03 | 5.49 e-03 |
| WP_PI3KAKT_SIGNALING_PATHWAY | 285 | 3.99 e-01 | 1.76 e+00 | 1.19 e-03 | 9.73 e-03 | 5.49 e-03 |

## High and low risk GSEA

Differential analysis, conducted using the R package limma, identified 113 DEGs between the HighRisk and LowRisk groups. Of these, 82 genes were up-regulated (logFC > 1 and adj.p < 0.05), and 31 were down-regulated (logFC < −1 and adj.p < 0.05).

A volcano plot was created to visualize these results (Fig 10A), and a heatmap displaying the top 10 up-regulated and down-regulated DEGs based on logFC was generated (Fig 10B).

GSEA was performed to explore the involvement of all expressed genes in biological processes, cellular components, and molecular functions. The results were visualized using a bubble chart (Fig 10C) and are summarized in Table 4.

GSEA indicated significant enrichment of genes in key pathways and functions (IL-6/JAK/STAT3 Signaling Pathway; TGF-β Pathway; Signaling by Hippo; Hedgehog/Gli Pathway), essential for understanding the molecular mechanisms of AP, and highlighted BP, CC, and MF significantly affected by gene expression changes in AP.

## Construction of control network

Next, using the genetic ChIPBase database and TF data for model genes, an mRNA-TF regulatory network was constructed, involving three model genes and 20 TFs (Fig 11A). Detailed information is provided in S1 Table.

Additionally, miRNAs related to the model genes were obtained from the StarBase database, and an mRNA-miRNA regulatory network was constructed and visualized using Cytoscape software (Fig 11B). This network includes four model genes and 25 miRNAs, with specific details listed in S2 Table.

## Immune infiltration analysis of AP (CIBERSORT)

The abundance of 22 immune cell types was estimated from the integrated GEO datasets. Immune infiltration analysis revealed the proportion of immune cells in the combined GEO datasets, with Fig 12A showing the immune cell

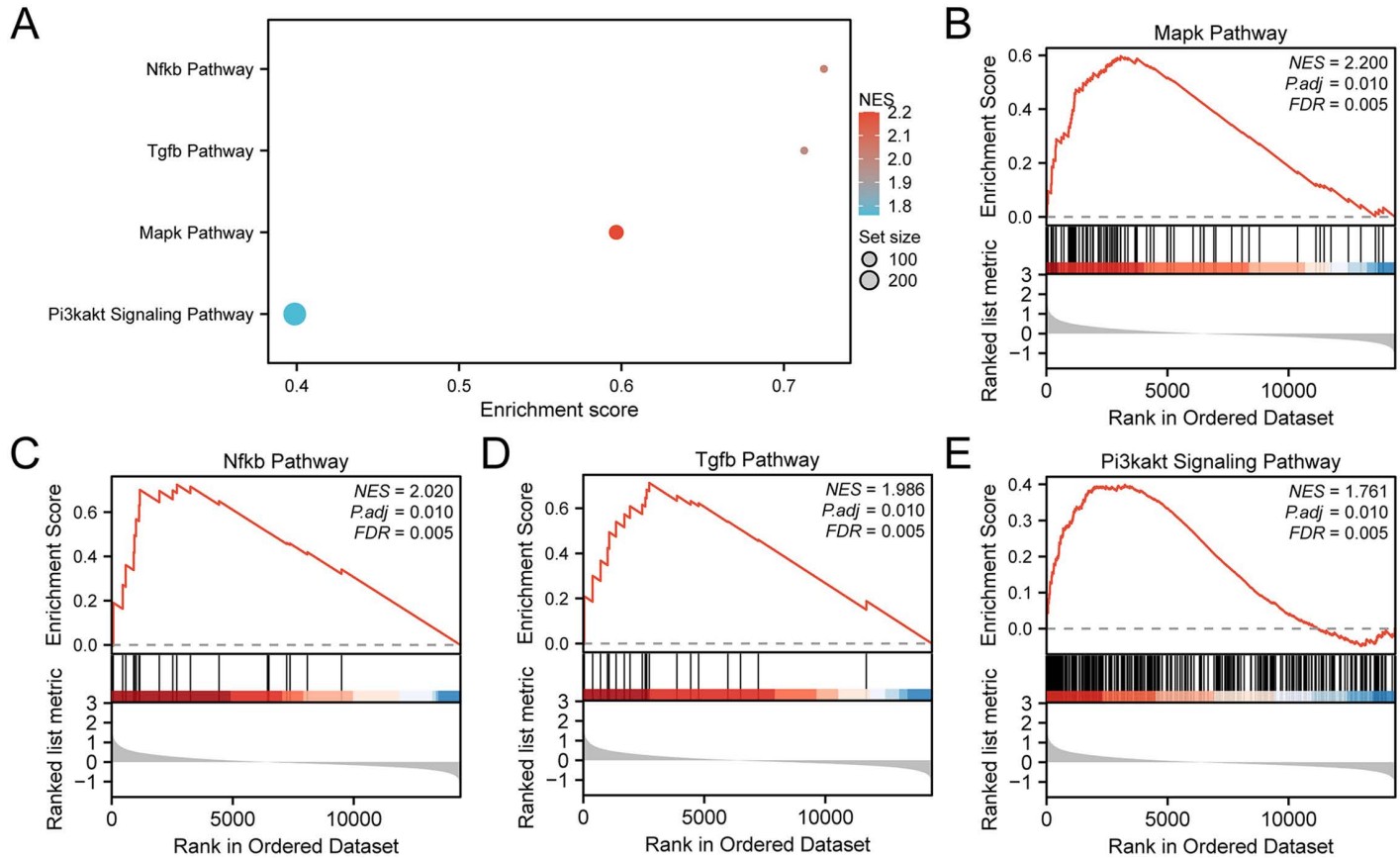

**Fig 6. GSEA for combined datasets.** (A) Bubble plot summarizing the significantly enriched pathways in the combined GEO datasets. (B–E) Enrichment plots for the MAPK pathway (B), NF-κB pathway (C), TGF-β pathway (D), and PI3K-Akt signaling pathway (E). In the bubble plot, bubble size represents gene count and color represents enrichment significance.

composition. A group comparison chart (Fig 12B) highlighted five immune cell types with significant differences (p < 0.05): M1 macrophages, Treg cells, CD4 memory T cells, Th17 cells, and monocytes.

A correlation heatmap (Fig 12C) was generated to show immune cell infiltration in AP samples. Most immune cells exhibited strong correlations, with M1 macrophages and monocytes showing the highest positive relevance.

Finally, a bubble chart (Fig 12D) was created to illustrate the relationship between model genes and immune cell infiltration abundance. This chart revealed that Iqgap1 was strongly correlated with most immune cells, with Treg cells exhibiting the strongest negative correlation.

## Immune infiltration analysis of High and Low Risk Groups (CIBERSORT)

The immune infiltration abundance of 22 immune cell types was calculated using the CIBERSORT algorithm, applied to AP samples from the combined GEO datasets. Based on the immune infiltration analysis, a group comparison chart was generated to highlight differences in immune cell infiltration between various groups. The analysis revealed significant differences in the abundance of eight immune cell types: CD8 memory T cells, M1 macrophages, M2 macrophages, Treg cells, Th1 cells, monocytes, gamma delta T cells, and activated NK cells (Fig 13A).

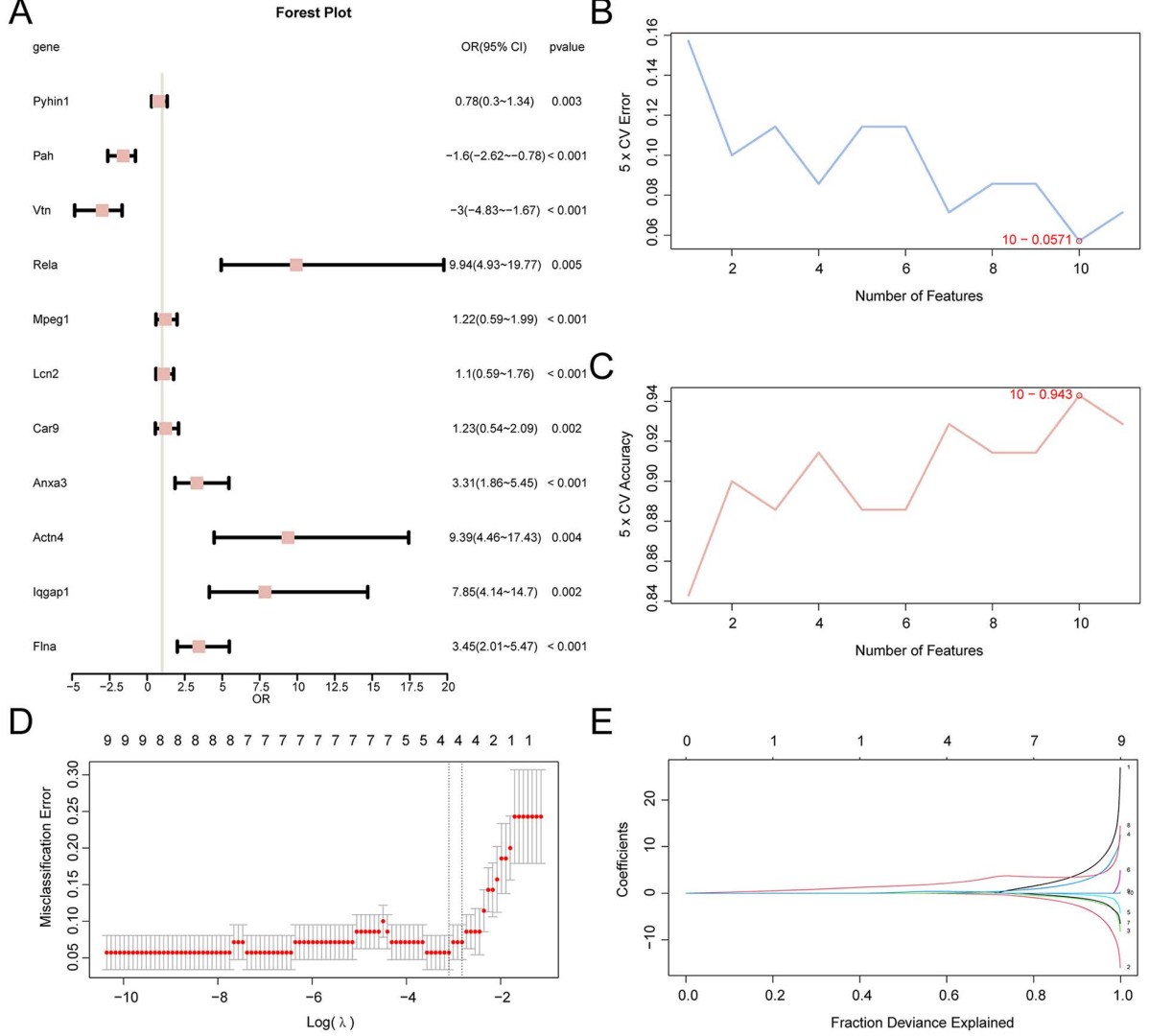

**Fig 7. Construction of the diagnostic model for AP.** (A) Forest plot of the 11 PRDEGs in the AP diagnostic model. (B, C) SVM model performance showing the number of genes associated with the lowest error rate (B) and highest accuracy (C). (D, E) LASSO regression model plot (D) and coefficient trajectory plot (E).

Correlation heatmaps were generated to further investigate immune infiltration in AP samples. Fig 13B-13C revealed that in LowRisk samples, most immune cells exhibited strong correlations, with Treg cells and monocytes showing the strongest negative correlation. In HighRisk samples, strong correlations were also observed, with M1 macrophages and gamma delta T cells demonstrating the most significant negative correlations.

Bubble charts were created to illustrate the relationship between model genes and immune cell infiltration abundance. The bubble charts indicated that in LowRisk samples, most immune cells were strongly correlated with Iqgap1, with Treg cells showing the strongest negative correlation. In HighRisk samples, the majority of immune cells were strongly correlated with ANXA3, with monocytes displaying the strongest positive correlation.

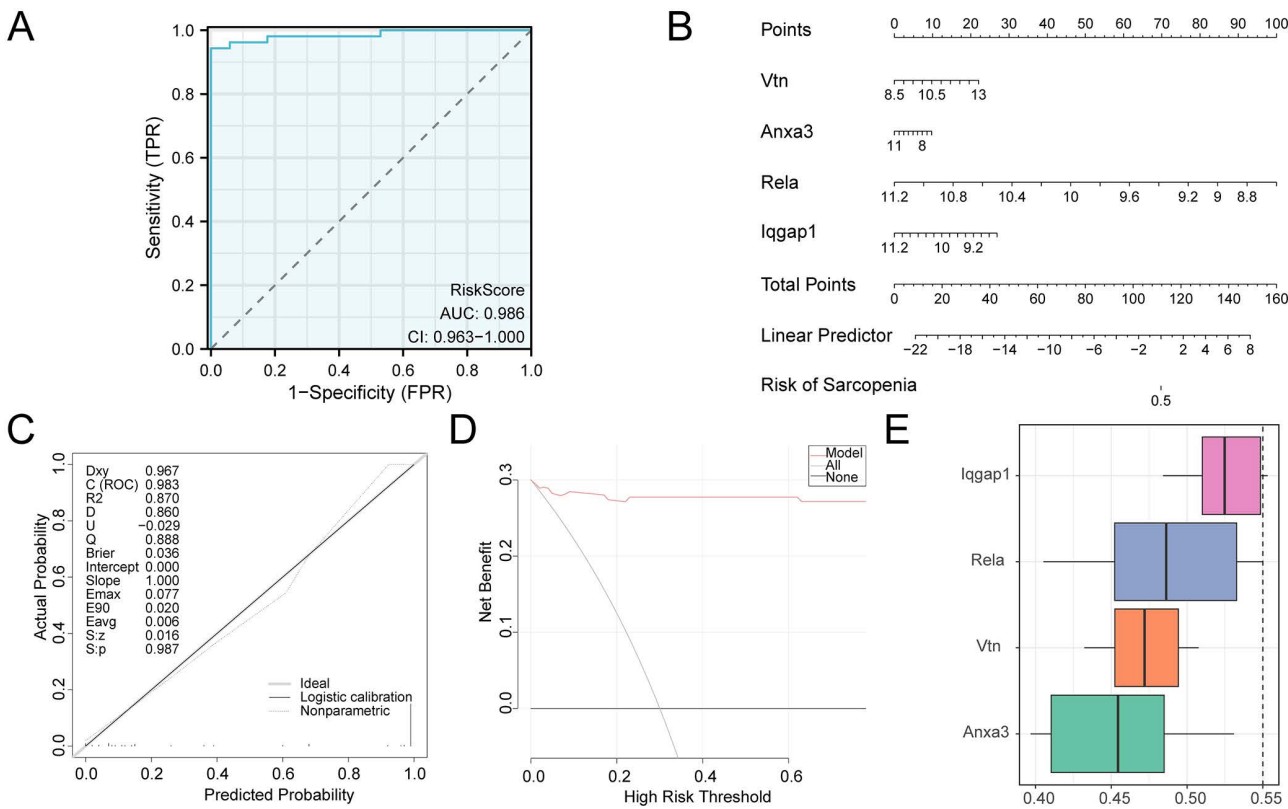

**Fig 8. Validation of the diagnostic model for AP and Friends analysis.** (A) ROC curve of the RiskScore in the combined datasets. (B) Nomogram of the AP diagnostic model based on the four model genes. (C) Calibration curve of the AP diagnostic model in the combined datasets. (D) DCA of the AP diagnostic model. (E) Friends scores of the four model genes, with the cut-off value indicated.

### External validation of the diagnostic model in the human RNA-seq cohort

To evaluate the potential of the four – gene model for the diagnosis of AP, we performed external validation using the independent human RNA-seq dataset GSE194331. An initial analysis of the 11 PRDEGs revealed that, despite the inherent interspecies transcriptional variability, most of the genes remained significantly dysregulated in human blood samples from AP patients (S1 Fig).

To assess the cross-species applicability of the discovery-phase signature, we first applied the four-gene model (ANXA3, IQGAP1, RELA, VTN) to an independent human blood cohort (GSE194331), using fixed coefficients derived from the murine dataset. However, this direct application resulted in poor discrimination, with an AUC of approximately 0.5. This finding underscores the limitations of direct model transferability and indicates considerable biological differences in gene expression profiles between species.

To address this limitation, we conducted an exploratory optimization. Differential expression analysis of the human cohort revealed that while ANXA3 and IQGAP1 consistently provided diagnostic signals (P<0.001; S1 Fig), RELA and VTN showed inconsistent trends and lacked statistical significance. Model comparison using the AIC further demonstrated that the two-gene model (ANXA3＋IQGAP1) offered a superior fit with lower complexity (AIC: 61.12) compared to the four-gene model (AIC: 63.01). Consequently, we recalibrated the diagnostic model by re-estimating the coefficients for ANXA3 and IQGAP1 using logistic regression in the human dataset.

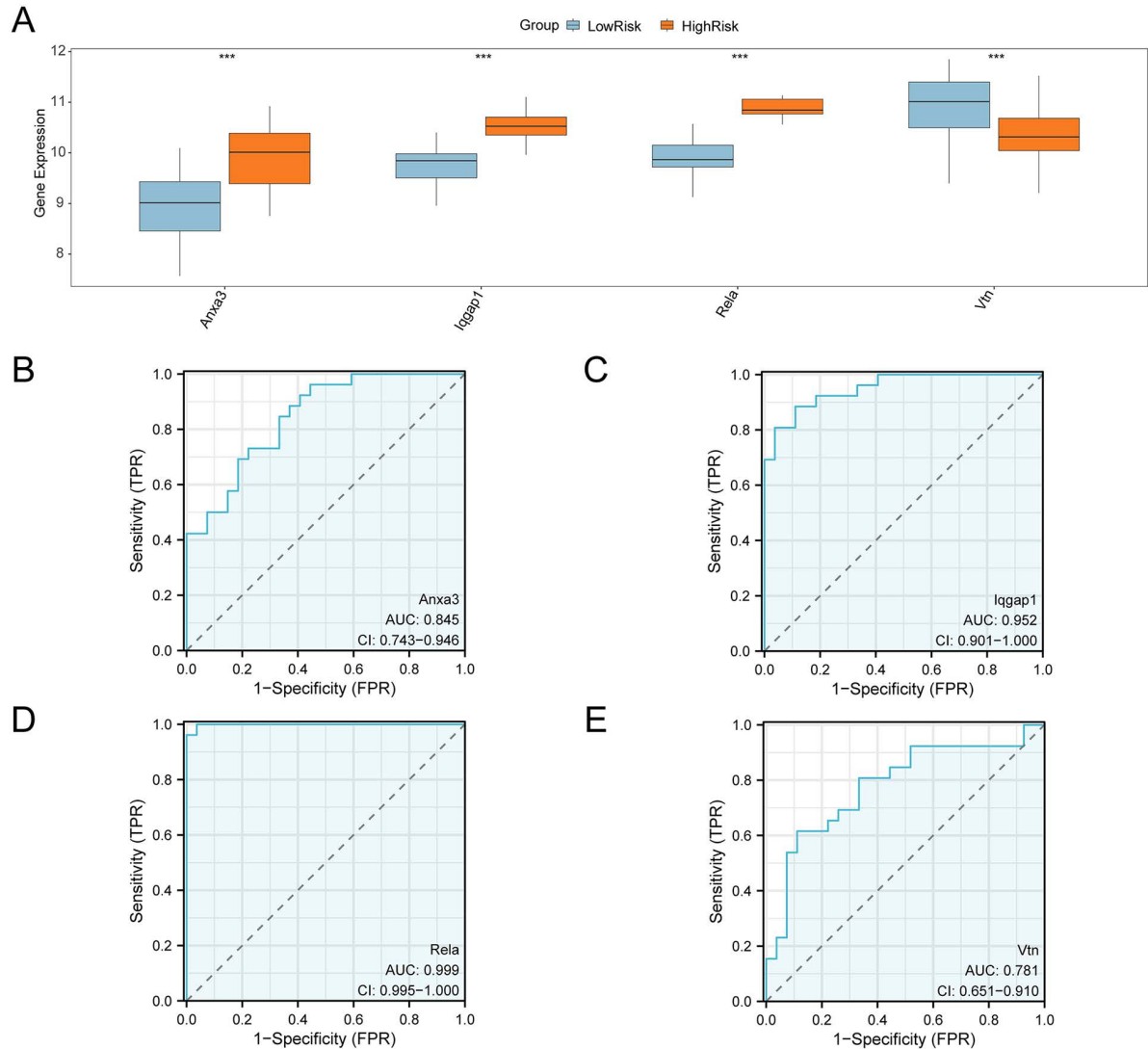

**Fig 9. Differential expression and ROC analysis of model genes in high-risk and low-risk AP groups.** (A(A) Boxplots showing the expression levels of ANXA3, IQGAP1, RELA, and VTN in the high-risk and low-risk groups. (B–E) ROC curves for ANXA3 (B), IQGAP1 (C), RELA (D), and VTN (E) in distinguishing the high-risk and low-risk groups.

Following recalibration, the optimized two-gene model exhibited a substantial restoration in performance, with an AUC of 0.957 (95% CI: 0.926–0.989) (Fig 14A; S5 Table). This notable improvement highlights the importance of incorporating species-specific expression weights to enhance diagnostic accuracy. Calibration analysis further confirmed the reliability of the recalibrated model. The calibration curve demonstrated strong alignment between predicted probabilities and observed outcomes (Fig 14C, 14D), as evidenced by a low Brier score of 0.0774 and a perfect calibration slope of 1.00 (S4 Table). Furthermore, DCA validated that the two-gene model delivers substantial net clinical benefit across a wide range of threshold probabilities, surpassing the performance of default strategies (Fig 14B).

In a secondary exploratory analysis of disease severity (Mild vs. Severe AP), the four-gene model demonstrated moderate discriminative power, with an AUC of 0.779 (S2B Fig). Notably, ANXA3 alone exhibited a stronger ability to differentiate between disease severity levels, as shown by the boxplots for different severity categories (S2A Fig). It also displayed

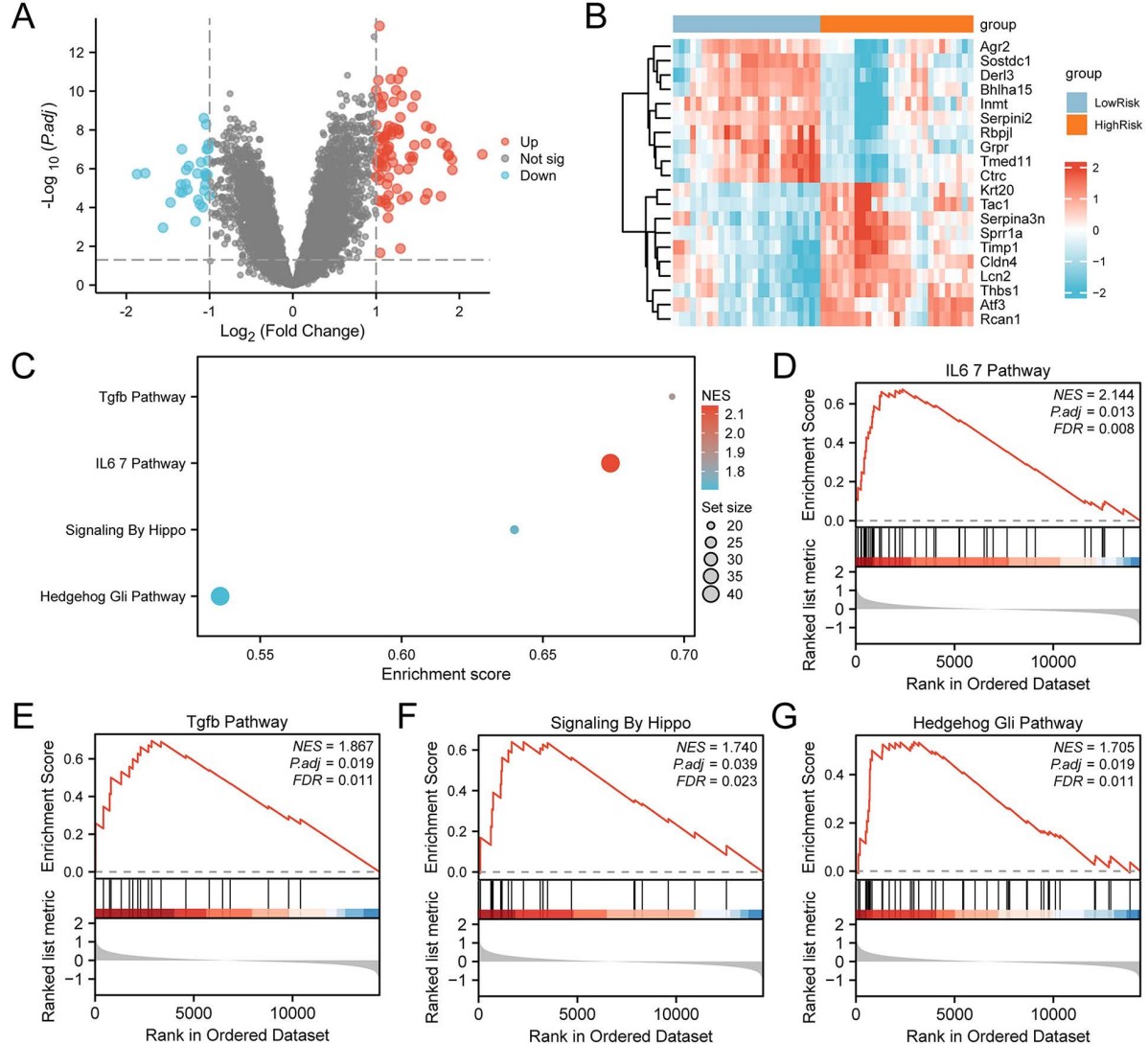

**Fig 10. Differential expression analysis and GSEA of high-risk and low-risk AP groups.** (A) Volcano plot of DEGs between the high-risk and low-risk groups. (B) Heatmap of the top 10 DEGs ranked by logFC. (C) Bubble plot summarizing the enriched pathways in AP samples. (D–G) Representative GSEA plots for the IL-6-related pathway (D), TGF-β pathway (E), Hippo signaling pathway (F), and Hedgehog/GLI pathway (G). NES, adjusted P value, and FDR are shown for each pathway.

comparable performance (AUC = 0.721) (S2C Fig), suggesting that while the primary utility of the signature lies in diagnosis, it may also provide modest insight into disease progression.

## Discussion

AP is an acute inflammatory condition with outcomes ranging from self-limited illness to life-threatening organ failure, contributing to significant morbidity and healthcare burdens, particularly in severe cases [28,29]. Clarifying the molecular mechanisms and developing reproducible diagnostic markers remain critical clinical priorities.

Principal findings: Three murine transcriptomic datasets were integrated to map the pyroptosis-related transcriptional landscape, identifying 11 PRDEGs associated with AP. Functional enrichment analysis revealed that these genes are

**Table 4. GSEA for risk groups.**

| ID | Set Size | Enrichment Score | NES | p value | p.adjust | q value |
|---|---|---|---|---|---|---|
| REACTOME_INTERLEUKIN_4_AND_INTERLEUKIN_13_SIGNALING | 92 | 6.69 e-01 | 2.44 e+00 | 1.54 e-03 | 1.32 e-02 | 7.96 e-03 |
| REACTOME_INTERLEUKIN_10_SIGNALING | 34 | 7.92 e-01 | 2.41 e+00 | 1.71 e-03 | 1.32 e-02 | 7.96 e-03 |
| KEGG_PATHOGENIC_ESCHERICHIA_COLI_INFECTION | 41 | 7.61 e-01 | 2.40 e+00 | 1.71 e-03 | 1.32 e-02 | 7.96 e-03 |
| WP_PATHOGENIC_ESCHERICHIA_COLI_INFECTION | 41 | 7.61 e-01 | 2.40 e+00 | 1.71 e-03 | 1.32 e-02 | 7.96 e-03 |
| REACTOME_TOLL_LIKE_RECEPTOR_CASCADES | 138 | 6.22 e-01 | 2.40 e+00 | 1.50 e-03 | 1.32 e-02 | 7.96 e-03 |
| REACTOME_TOLL_LIKE_RECEPTOR_TLR1_TLR2_CASCADE | 102 | 6.50 e-01 | 2.39 e+00 | 1.55 e-03 | 1.32 e-02 | 7.96 e-03 |
| REACTOME_FORMATION_OF_THE_CORNIFIED_ENVELOPE | 66 | 6.92 e-01 | 2.39 e+00 | 1.60 e-03 | 1.32 e-02 | 7.96 e-03 |
| WP_INTEGRINMEDIATED_CELL_ADHESION | 87 | 6.59 e-01 | 2.38 e+00 | 1.54 e-03 | 1.32 e-02 | 7.96 e-03 |
| REACTOME_SIGNALING_BY_INTERLEUKINS | 373 | 5.46 e-01 | 2.36 e+00 | 1.31 e-03 | 1.32 e-02 | 7.96 e-03 |
| PID_PDGFRB_PATHWAY | 115 | 6.21 e-01 | 2.34 e+00 | 1.52 e-03 | 1.32 e-02 | 7.96 e-03 |
| WP_NETWORK_MAP_OF_SARSCOV2_SIGNALING_PATHWAY | 173 | 5.84 e-01 | 2.33 e+00 | 1.46 e-03 | 1.32 e-02 | 7.96 e-03 |
| REACTOME_RHO_GTPASES_ACTIVATE_FORMINS | 104 | 6.24 e-01 | 2.31 e+00 | 1.55 e-03 | 1.32 e-02 | 7.96 e-03 |
| WP_FIBRIN_COMPLEMENT_RECEPTOR_3_SIGNALING_PATHWAY | 39 | 7.38 e-01 | 2.30 e+00 | 1.70 e-03 | 1.32 e-02 | 7.96 e-03 |
| WP_TGFBETA_SIGNALING_PATHWAY | 122 | 5.99 e-01 | 2.29 e+00 | 1.49 e-03 | 1.32 e-02 | 7.96 e-03 |
| WP_SPINAL_CORD_INJURY | 98 | 6.24 e-01 | 2.28 e+00 | 1.56 e-03 | 1.32 e-02 | 7.96 e-03 |
| KEGG_LEUKOCYTE_TRANSENDOTHELIAL_MIGRATION | 99 | 6.21 e-01 | 2.28 e+00 | 1.55 e-03 | 1.32 e-02 | 7.96 e-03 |
| PID_IL6_7_PATHWAY | 44 | 6.74 e-01 | 2.14 e+00 | 1.72 e-03 | 1.32 e-02 | 7.96 e-03 |
| BIOCARTA_TGFB_PATHWAY | 19 | 6.96 e-01 | 1.87 e+00 | 3.55 e-03 | 1.89 e-02 | 1.14 e-02 |
| REACTOME_SIGNALING_BY_HIPPO | 20 | 6.40 e-01 | 1.74 e+00 | 1.05 e-02 | 3.88 e-02 | 2.35 e-02 |
| PID_HEDGEHOG_GLI_PATHWAY | 44 | 5.36 e-01 | 1.70 e+00 | 3.44 e-03 | 1.89 e-02 | 1.14 e-02 |

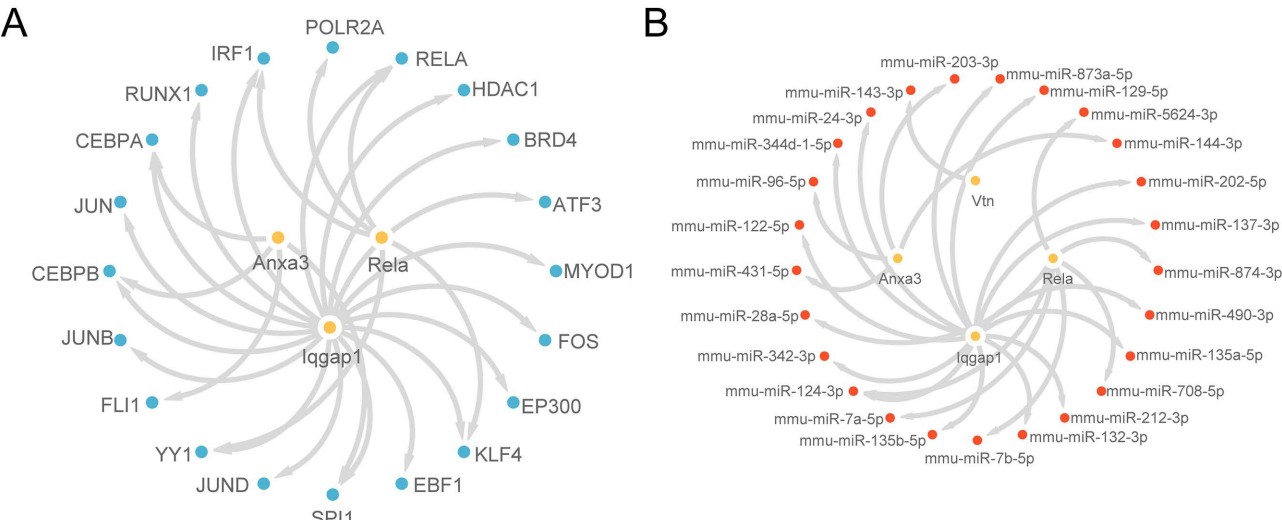

**Fig 11. Regulatory networks of the model genes.** (A) mRNA–TF regulatory network of the model genes.(B) mRNA–miRNA regulatory network of the model genes.

involved in cytoskeletal organization and key inflammatory pathways, including MAPK and NF-κB. Immune infiltration analysis further highlighted a distinct immune microenvironment in AP, characterized by an increased presence of M1 macrophages and monocytes, while Iqgap1 was strongly negatively correlated with Treg cell abundance. Based on these

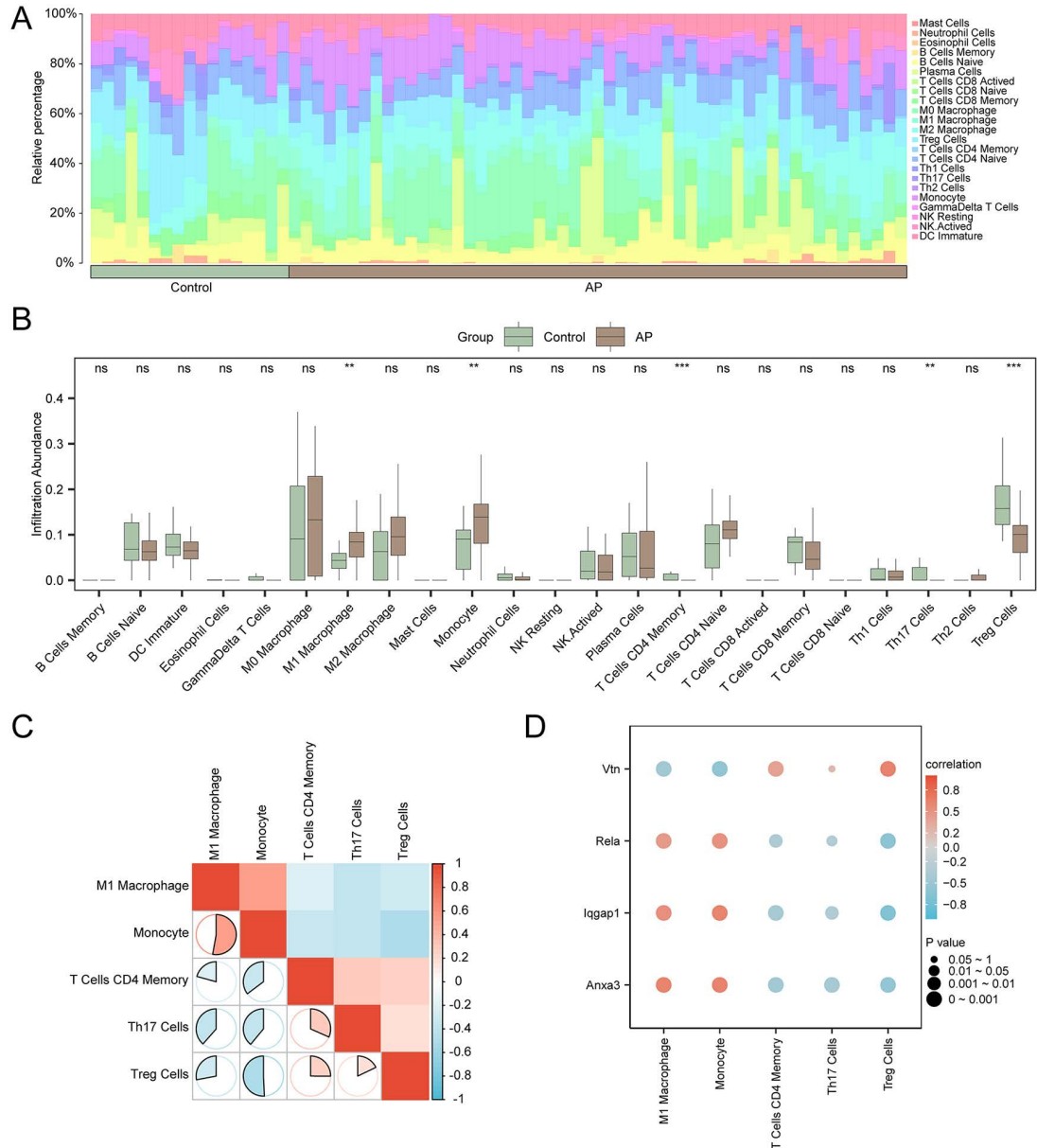

**Fig 12. Combined datasets immune infiltration analysis by CIBERSORT algorithm.** A-B. Immune cell composition in the combined datasets, shown as a histogram (A) and a boxplot comparing groups (B). C. Heatmap of correlations between immune cell types in the combined GEO datasets. D. Correlation bubble plot showing the relationship between immune cell infiltration abundance and model genes. Statistical significance is indicated as: ** p < 0.01, *** p < 0.001. Green represents the Control group, and brown represents the AP group. Red indicates a positive correlation, while blue indicates a negative correlation.

findings, a diagnostic model comprising four candidate genes (ANXA3, IQGAP1, RELA, and VTN) was initially constructed in the murine discovery cohort. Risk stratification based on this model effectively classified patients into high- and low-risk groups. GSEA revealed significant enrichment of IL-6/JAK/STAT3 and TGF-β signaling pathways in high-risk samples, indicating a more severe inflammatory state. However, considering the complexity of cross-species translation, we performed rigorous model optimization in the human blood cohort. Although RELA and VTN were significant in murine

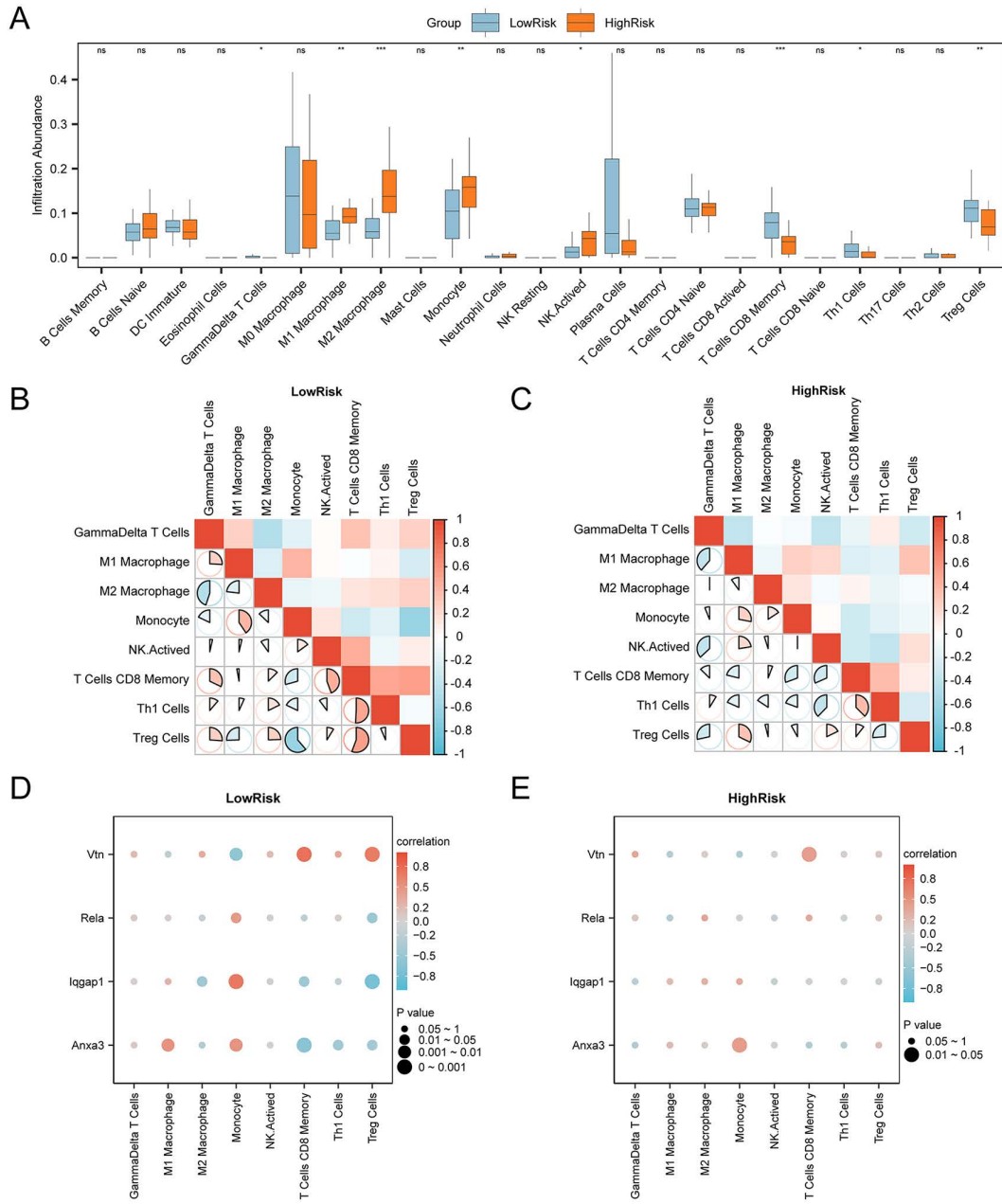

**Fig 13. Risk group immune infiltration analysis by CIBERSORT algorithm.** A A. Comparison of immune cell infiltration between the LowRisk and HighRisk groups in AP samples. B-C. Heatmaps showing the correlation between immune cell types in AP samples for the LowRisk (B) and HighRisk (C) groups. D-E. Bubble plots showing the correlation between immune cell infiltration abundance and model genes in the LowRisk (D) and HighRisk (E) groups. Statistical significance is indicated as: * p < 0.05, ** p < 0.01, *** p < 0.001. Blue represents the LowRisk group, and orange represents the HighRisk group. Red indicates a positive correlation, and blue indicates a negative correlation.

pancreas tissue, they exhibited inconsistent expression in human blood. Consequently, using the AIC, the panel was refined into a robust two-gene diagnostic model (ANXA3 and IQGAP1). In the independent human cohort, this recalibrated signature maintained superior accuracy with an AUC of 0.957. Its robustness was further validated by excellent calibration

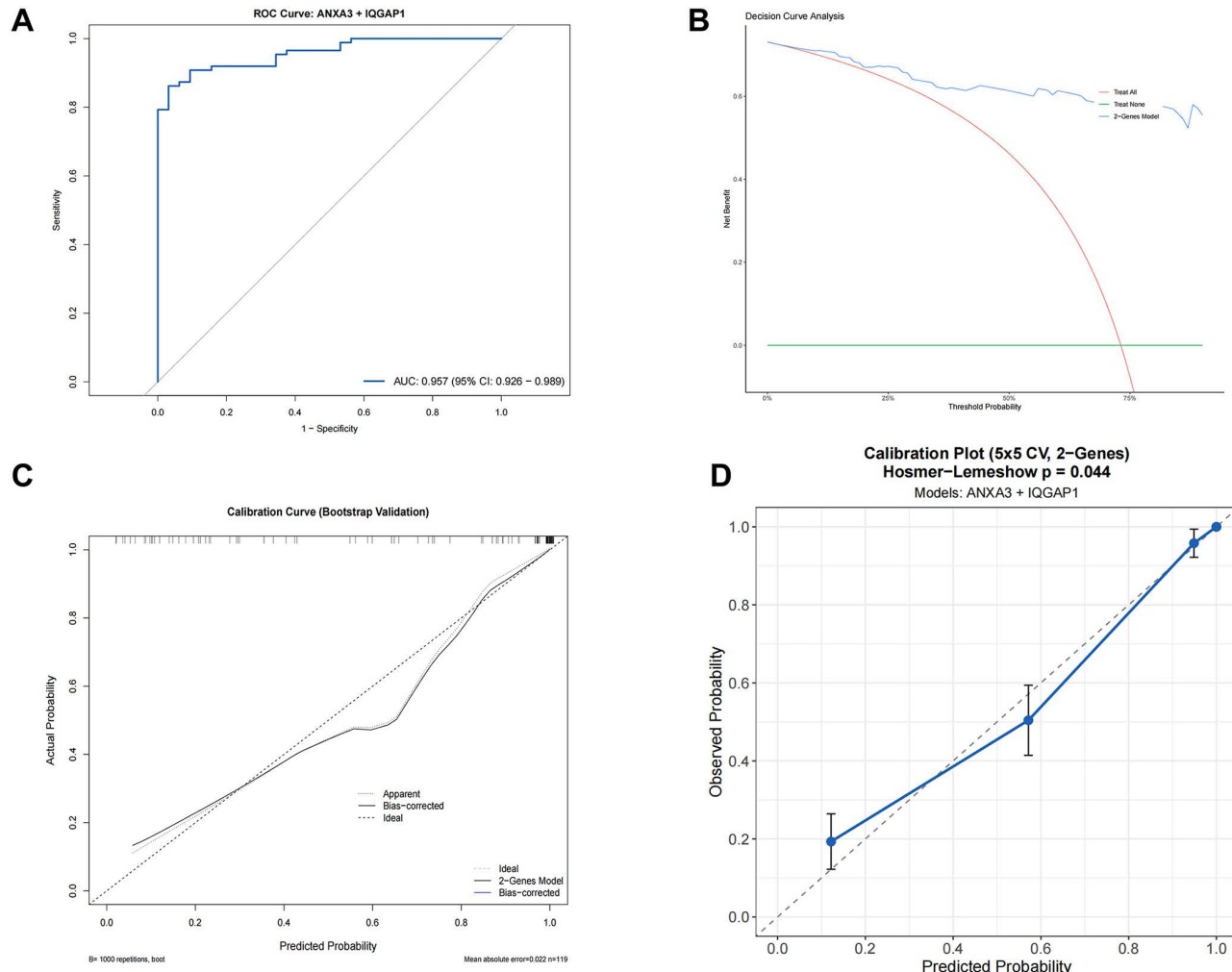

**Fig 14. Validation and performance assessment of the recalibrated two-gene diagnostic model in the independent human cohort.** (A) ROC curve showing the diagnostic performance of the recalibrated two-gene signature (ANXA3 + IQGAP1) in the human peripheral blood cohort. The model achieved an AUC of 0.957 (95% CI: 0.926–0.989). (B) DCA showing the clinical net benefit of the two-gene model across different threshold probabilities, compared with the treat-all and treat-none strategies. (C) Bootstrap calibration curve based on 1,000 resamples, demonstrating the agreement between predicted and observed probabilities. (D) Calibration plot showing the relationship between predicted and observed probabilities, with the Hosmer–Lemeshow test result indicated.

(Brier score: 0.0774) and DCA, while secondary analyses highlighted the potential utility of ANXA3 alone in stratifying disease severity. These findings establish the two-gene signature as a precise, clinically interpretable, and highly translatable tool for AP diagnosis. Although the transition from murine pancreatic tissue to human peripheral blood represents a shift in biological domains, our results suggest that circulating leukocyte transcriptomes reliably reflect the local inflammatory state [30]. This indicates that the immune signature initiated in the pancreas is effectively captured in the systemic circulation, providing a non-invasive method to monitor organ-specific injury.

Biological plausibility and relation to prior work: Pyroptosis has been implicated in various inflammatory conditions [31]. The prominence of RELA, IQGAP1, ACTN4, FLNA, and ANXA3 aligns with their known roles in cytokine induction, cytoskeletal dynamics, and injury responses [32–36]. Specifically, ANXA3 is recognized as a distinct marker of neutrophil

granules. Its substantial upregulation in peripheral blood suggests that the model effectively captures the extensive mobilization and activation of neutrophils, a hallmark of the early systemic inflammatory response in AP [37]. Meanwhile, VTN is involved in adhesion and complement regulation [38]. Other PRDEGs are linked to complementary processes: LCN2 and MPEG1 to innate immunity, PAH to amino acid metabolism, CA9 to acid–base homeostasis, and PYHIN1 to inflammatory signaling [39–43]. Consistent with these gene functions, enrichment analysis emphasized cell–substrate adhesion and amoeboid-type migration, processes critical to leukocyte trafficking and tissue remodeling in AP [44–46]. The immune deconvolution findings corroborate previous observations of macrophages, neutrophils, and T-cell subsets in AP [47–51], providing mechanistic insight into the PRDEG panel [52].

Clinical Implications: This study provides a molecular framework for incorporating pyroptosis into the diagnostic and mechanistic evaluation of AP. By identifying specific pyroptosis-related signatures, it offers an alternative approach for early diagnosis, potentially serving as a complementary tool when conventional biochemical markers are inconclusive. The finding that ANXA3 alone demonstrates significant discriminative power in differentiating severe AP suggests its dual potential as both a diagnostic marker and an early indicator of disease severity. This could aid clinicians in rapid risk stratification and triage. Furthermore, the correlation between IQGAP1 and immune tolerance, specifically Treg abundance, provides valuable mechanistic insights into the immune-inflammatory state of patients, offering a foundation for future investigations into immunomodulatory therapies targeting pyroptosis pathways.

Strengths and Limitations: This study has several notable strengths. First, multiple independent transcriptomic datasets were integrated, and batch effects were systematically controlled, enhancing the robustness of the findings. Second, by focusing on pyroptosis-related pathways, a clear and interpretable gene diagnostic model was developed. Third, comprehensive external validation was performed in an independent human cohort, where the model's discriminative performance, calibration accuracy, and clinical net benefit were assessed, with full transparency regarding model coefficients and per-sample predictions. Several limitations should be considered. The most significant is the biological domain shift from murine pancreatic tissue in the discovery phase to human peripheral blood in the validation phase. This transition means that, while the murine model focused on localized pancreatic injury, the human validation phase emphasized systemic inflammatory signals. Our analysis identified tissue-specific markers, such as RELA and VTN, that did not translate well to blood-based diagnostics, requiring recalibration of the model. Although this optimization reduced species-specific expression differences, biological biases due to sample-type variation could not be fully eliminated. Moreover, while the two-gene model demonstrated excellent diagnostic accuracy, its ability to stratify disease severity was primarily driven by ANXA3, suggesting that different components of the signature may have distinct clinical implications. Furthermore, as a retrospective study with a moderate sample size, residual confounding factors—such as clinical comorbidities or unmeasured covariates—may have influenced the results. While immune cell deconvolution provided valuable insights, it relied on algorithmic assumptions and lacked direct flow cytometric validation. Finally, the absence of prospective multi-center validation and targeted functional experiments (such as in vitro gene knockdown) represents a significant gap. Future research should focus on investigating the mechanistic roles of ANXA3 and IQGAP1 in AP-driven pyroptosis and validating the signature in larger, prospective clinical cohorts.

This study defines a reproducible pyroptosis-related transcriptional signature for AP and establishes a streamlined two-gene diagnostic model (ANXA3 and IQGAP1) with demonstrated cross-species translatability. Initially derived from murine pancreatic tissue, the recalibrated version of this signature maintained exceptional diagnostic performance in human peripheral blood, highlighting its potential as a reliable biomarker. Despite its high accuracy, the transition from localized tissue signals to systemic circulation requires careful biological interpretation. Future research will focus on: (i) biological validation through protein-level confirmation and assessment of tissue–blood expression concordance; (ii) specificity testing against other acute inflammatory conditions (such as sepsis or bacteremia) to verify the pancreas-specific diagnostic relevance; (iii) prospective multi-center clinical evaluation to further validate its utility in early severity stratification, particularly the independent predictive value of ANXA3, and its incremental benefit beyond conventional markers; and (iv)

the development of standardized clinical assays, such as RT-qPCR, to facilitate robust and reproducible translation into emergency clinical settings.

## Supporting information

**S1 Fig. Boxplot of 11 PRDEGs expression across groups.** Boxplots illustrating the relative expression levels of 11 pyroptosis-related differentially expressed genes (PRDEGs)—RELA, IQGAP1, ACTN4, FLNA, ANXA3, VTN, LCN2, MPEG1, PAH, CA9 (homologous to murine Car9), and PYHIN1—were generated within the independent human peripheral blood dataset. This analysis assesses the cross-species translatability of the initial murine-derived signature by comparing the differential expression patterns of these candidate genes between human patients with AP and healthy controls. (TIF)

**S2 Fig. Association of signature genes with disease severity and their predictive performance in the human blood cohort.** The figure depicts the relationship between gene expression and the severity of acute pancreatitis (Mild, Moderate, and Severe) using a human peripheral blood dataset. (A) Boxplots show the relative expression levels of the signature genes across the three severity groups, emphasizing significant differential expression as the disease progresses. (B-C) ROC curves assess the discriminatory ability (AUC) of the multi-gene panel compared to ANXA3 alone in predicting disease severity. (TIFF)

**S3 Fig. Diagnostic performance comparison between the initial four-gene and optimized two-gene models.** The figure illustrates the diagnostic performance, calibration, and clinical utility of the initial four-gene signature (ANXA3, IQGAP1, RELA, and VTN) compared to the AIC-optimized two-gene signature (ANXA3 and IQGAP1) in the human validation cohort. Performance is evaluated through ROC curves to assess discriminatory accuracy (AUC) (A), calibration plots (B) to evaluate the goodness-of-fit between predicted and observed outcomes, and (DCA) (C) to determine clinical net benefit across various threshold probabilities. (TIF)

**S4 Fig. Split-sample validation and diagnostic performance of the recalibrated two-gene model.** The ROC curve was used to assess the discriminatory accuracy of the optimized two-gene signature (ANXA3 and IQGAP1) in an independent human testing set. To mitigate the risk of overfitting, the external human cohort (GSE194331, N = 119) was randomly divided into a training set (50%, n = 60) for coefficient re-estimation and an independent testing set (50%, n = 59). The curve illustrates the model's performance when applied exclusively to the testing set, demonstrating that the high diagnostic accuracy is robust and not a result of overfitting. (TIFF)

**S1 Table. Predicted TF regulatory network for the four model genes.** This table lists the potential TF associated with the identified four model genes. (CSV)

**S2 Table. Predicted miRNA–mRNA regulatory network of the four model genes.** This table details the miRNAs targeted to the four model genes. (CSV)

**S3 Table. Detailed logistic regression parameters and odds ratios for the optimized two-gene diagnostic model.** This table presents the statistical summary of the refined diagnostic model in the human validation cohort. It includes the estimated coefficients, standard errors, z-values, and p-values. Odds ratios (OR) with their corresponding 95% CI are provided to quantify the association between the expression levels of individual genes and the likelihood of AP. (CSV)

**S4 Table. Recalibration metrics and Brier score for the two-gene model.** To assess the concordance between predicted risks and observed outcomes in the human cohort, this table provides key calibration metrics, including the calibration intercept (calibration-in-the-large), calibration slope, and the Brier score (a measure of overall prediction accuracy). These metrics indicate the goodness-of-fit and the reliability of the model's probability estimates.
(CSV)

**S5 Table. AUC for the two-gene signature.** The table summarizes the diagnostic discrimination performance of the combined ANXA3 and IQGAP1 signature. The AUC value is provided along with its 95% CI to illustrate the model's ability to accurately distinguish AP patients from healthy controls.
(CSV)

**S6 Table. Performance metrics from 5-fold repeated cross-validation (5x5 CV).** This table summarizes the stability and robustness of the two-gene model through repeated cross-validation (5 folds, 5 repeats). It includes the mean values and standard deviations (SD) for the area under the ROC curve (ROC), sensitivity (Sens), and specificity (Spec), demonstrating that the model's performance is consistent and not biased by a specific data split.
(CSV)

**S7 Table. Internal validation and optimism correction of the initial four-gene diagnostic model.** This table details the rigorous internal validation metrics for the initial four-gene signature (Anxa3, Iqgap1, Rela, and Vtn) based on 500 bootstrap iterations. It compares the "Apparent" performance (measured on the original dataset) with the "Bootstrap-corrected" performance to estimate "Optimism." Metrics include the AUC, Brier score, calibration intercept, and calibration slope. The Optimism-corrected values provide a more realistic and conservative estimate of the model's predictive performance by accounting for potential over-fitting during the initial model construction phase.
(CSV)

**S8 Table. Bootstrap-based selection frequencies of the 11 PRDEGs in the diagnostic model.** To evaluate the stability of the feature selection process, this table presents the frequency with which each of the 11 initial PRDEGs was selected across 500 bootstrap iterations. Genes with higher selection frequencies (Such as Anxa3 at 0.91) represent the most robust and reliable predictors for the diagnostic signature.
(CSV)

## Author contributions

**Conceptualization:** Yun Pan.

**Formal analysis:** Zhongsu Yu, Shuyuan Li, Liangping Cheng.

**Funding acquisition:** Guangyuan Yu.

**Investigation:** Yuxia Chen, Yun Pan, Liangping Cheng.

**Methodology:** Zhongsu Yu, Shuyuan Li, Yuxia Chen.

**Writing – original draft:** Yuting Wang, Jun Li.

**Writing – review & editing:** Guangyuan Yu.

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
