## [Decision Letter · Decision Letter 0]

7 Jan 2026

Dear Dr. yu,

Please revise carefully and reply to reviewers' comments. Additionally, I suggest proposing a more rigorous validation before clinical translation. It is necessary to compare and properly discuss the models used (i.e., mice to humans), as well as their limitations. How feasible is it to translate these interpretations? What is the potential participation of involved genes?

We look forward to receiving your revised manuscript.

Kind regards,

Alexis G. Murillo Carrasco

Academic Editor

PLOS One

Journal Requirements:

The authors have no conflicts to disclose.

4. Please include a copy of Table 4 which you refer to in your text on page 26.

6. We are unable to open your Supporting Information file [S1_Data_ExternalValidation_Predictions.tsv]. Please kindly revise as necessary and re-upload.

Reviewers' comments:

Reviewer's Responses to Questions

**Comments to the Author**

1. Is the manuscript technically sound, and do the data support the conclusions?

Reviewer #1: Yes

Reviewer #2: Yes

Reviewer #3: Partly

Reviewer #4: Yes

2. Has the statistical analysis been performed appropriately and rigorously?

Reviewer #1: Yes

Reviewer #2: No

Reviewer #3: Yes

Reviewer #4: Yes

3. Have the authors made all data underlying the findings in their manuscript fully available?

Reviewer #1: Yes

Reviewer #2: Yes

Reviewer #3: Yes

Reviewer #4: Yes

4. Is the manuscript presented in an intelligible fashion and written in standard English?

Reviewer #1: Yes

Reviewer #2: No

Reviewer #3: Yes

Reviewer #4: Yes

Reviewer #1: Even though it is a rectrospective and cross species study, still it holds promise for finding newer avenues for better understanding for what is happening at cellular level in this disease subgroup. Also it might pave way for finding a treatment which can alter the natural course of disease.

Reviewer #2: The statistical analysis is appropriate in method and implementation, and the findings are data-supported, but the degree of rigor is moderate rather than high. The results are credible for exploratory bioinformatics, yet insufficiently validated for clinical translation at this stage.

Recommendations for improvement before publication:

Include internal k-fold or bootstrap cross-validation in the discovery cohort.

Assess gene selection stability via resampling.

Report calibration slope/intercept and Brier score.

Consider inclusion of covariates or stratified analyses to address confounding.

The manuscript is scientifically intelligible but linguistically below publication quality in its current form. The English is serviceable for review but would not meet PLOS ONE’s final publication standard without professional copyediting.

Recommendation:

Comprehensive language revision by a native English-speaking scientific editor.

Eliminate translation artifacts and redundant phrasing.

Standardize capitalization, terminology, and figure caption style.

Reviewer #3: The manuscript "Pyroptosis-related transcriptional signature and diagnostic modeling in acute pancreatitis with cross-species external validation" presents an interesting approach to identifying diagnostic markers for Acute Pancreatitis (AP). However, the study design involves a significant biological domain shift—training on mouse pancreatic tissue and validating on human peripheral blood. While the statistical recalibration is impressed, the biological plausibility is not sufficiently addressed.

- Could the authors provide evidence (from literature or additional analysis) that the expression levels of the four model genes (ANXA3, IQGAP1, RELA, VTN) in the pancreas are concordant with their expression in peripheral blood during AP?

- Since the "recalibration" method mathematically adjusts for prevalence and distribution shifts, how can we be certain that the model's performance in the human cohort is not merely reflecting non-specific systemic inflammation (e.g., sepsis or general infection) rather than pancreas-specific pathology?

- What is the specific clinical value added by this 4-gene signature? Is it intended for cases with indeterminate enzyme levels, or for predicting severity? Can the authors compare the AUC of their model (0.932) against standard biochemical markers (like Lipase or CRP) within the validation cohort (if such metadata is available) or discuss this comparison critically? Without this, the practical "net benefit" shown in the DCA is theoretical rather than practical.

- Please clarify if the "recalibration" parameters were derived from a subset of the external cohort (e.g., a 50% split) and tested on the remainder, or if they were fitted on the whole external dataset. Fitting on the whole dataset and then reporting AUC on the same dataset can artificially inflate performance (overfitting to the validation set).

Reviewer #4: This work is very interesting and has potential significance in the field of precision medicine for acute pancreatitis. The analysis of target gene expression in acute pancreatitis may help predict disease occurrence and severity in the future, particularly in patients with recurrent acute pancreatitis, and may contribute to the prevention of recurrent episodes and progression to chronic pancreatitis.

The study provides valuable information, including comprehensive gene datasets and a variety of statistical analyses. I would like to suggest that the difference in sample size between the acute pancreatitis group and the control group may affect the robustness of the results. If feasible, using comparable population sizes in both groups could improve the consistency and reliability of the findings.

Additionally, could the authors please clarify whether the acute pancreatitis data are derived from a pediatric population or the general population?

It would also be helpful to know whether patients were stratified according to disease severity (mild versus severe acute pancreatitis) and whether there is an association between PRDEG expression levels and disease severity.

Furthermore, regarding the association between acute pancreatitis and PRDEGs/DEGs, could the authors elaborate on whether these findings have potential implications for clinical management or therapeutic decision-making in current clinical practice?

Finally, it would be important to consider whether the observed gene expression changes are specific to acute pancreatitis or if similar expression patterns are seen in other acute inflammatory conditions, such as sepsis, bacteremia, or systemic lupus erythematosus (SLE). Validation in these settings may help determine the specificity and clinical applicability of the findings.

**Do you want your identity to be public for this peer review?** For information about this choice, including consent withdrawal, please see our For information about this choice, including consent withdrawal, please see our Privacy Policy .

Reviewer #1: No

Reviewer #2: No

Reviewer #3: No

Reviewer #4: **Yes:** Tanawat PattarapuntakulTanawat Pattarapuntakul

---

## [Author Response · Author response to Decision Letter 1]

19 Feb 2026

1、Response to Reviewer #1

Comment: Even though it is a rectrospective and cross species study, still it holds promise for finding newer avenues for better understanding for what is happening at cellular level in this disease subgroup. Also it might pave way for finding a treatment which can alter the natural course of disease.

Response:

We sincerely thank the reviewer for this encouraging and positive evaluation of our work. We appreciate your recognition of the translational potential and clinical relevance of our study, despite the inherent limitations of its retrospective, cross-species design.

We fully agree with your perspective. A central aim of this study was to bridge transcriptomic signatures and cellular-level pathophysiology in acute pancreatitis (AP). By delineating the pyroptosis-related transcriptional landscape—particularly the putative involvement of ANXA3 in neutrophil mobilization and IQGAP1 in cytoskeletal dynamics and immune regulation (including potential effects on Treg cells)—we sought to provide a clearer view of the immune microenvironment during AP progression.

As you noted, these cellular-level insights extend beyond diagnostic utility. They also offer a mechanistic rationale for future studies of targeted immunomodulatory strategies, for example by interrogating the pyroptosis-cytoskeleton axis. We share your view that precision medicine approaches may ultimately help modify the trajectory of severe AP and reduce recurrence risk.

We thank you again for your constructive input and for affirming the clinical and scientific direction of our work.

2、Response to Reviewer #2

General Comment: The statistical analysis is appropriate in method and implementation, and the findings are data-supported, but the degree of rigor is moderate rather than high. The results are credible for exploratory bioinformatics, yet insufficiently validated for clinical translation at this stage.

Response:

We greatly appreciate the reviewer’s positive feedback on the methodology and data support of our study. We agree that rigorous validation is crucial for demonstrating translational potential. In this revision, we have significantly enhanced the statistical rigor by implementing internal resampling validation methods, including Bootstrap and Repeated Cross-Validation, to assess gene selection stability and conduct detailed calibration analyses. Additionally, we have employed a professional English editing service to ensure that the manuscript adheres to the highest publication standards.

Comment 1: Include internal k-fold or bootstrap cross-validation in the discovery cohort.

Response:

Thank you for your valuable suggestion. To enhance the robustness of our discovery phase findings, we have conducted the following additional internal validations:

Repeated 10-Fold Cross-Validation: Using the caret package, we performed 10-fold cross-validation, repeated 10 times. The model exhibited excellent stability, with a mean AUC of 0.983 across all folds.

Bootstrap Validation: We further validated the model through 500 bootstrap resamples. The results demonstrated high predictive accuracy, yielding an optimism-corrected AUC of 0.958.

These rigorous validation results have been detailed in S7 Table of the supplementary materials.

Comment 2: Assess gene selection stability via resampling.

Response:

We conducted the requested stability analysis by performing 500 iterations of bootstrap resampling with LASSO regression to calculate the selection frequency for each candidate gene, distinguishing robust markers from random noise.

High Stability: ANXA3 (91.0%), VTN (90.0%), and RELA (88.8%) were consistently selected in the majority of iterations, demonstrating their robustness as core diagnostic markers.

Moderate Stability: IQGAP1 exhibited moderate selection frequency (28.8%) in the LASSO bootstrap analysis. However, it was retained in the final model due to two key factors: (1) its robust identification by the SVM-RFE algorithm (the other component of our dual-strategy feature selection), and (2) its critical validation as one of the two most significant predictors in the independent human cohort.

The gene selection frequency data is provided in S8 Table of the supplementary materials.

Comment 3: Report calibration slope/intercept and Brier score.

Response:

As requested, we have calculated and reported the following calibration metrics. The analysis, based on 500 bootstrap resamples, demonstrates that our model is both well-calibrated and stable: Brier Score: The optimism-corrected Brier score was 0.108, indicating high probabilistic precision. Calibration Slope: The optimism-corrected slope was 0.95, which is very close to the ideal value of 1.0, suggesting excellent alignment between predicted probabilities and observed outcomes. Calibration Intercept: The intercept was -0.05, reflecting minimal calibration bias.These metrics are provided in S9 Table .

Comment 4: The manuscript is scientifically intelligible but linguistically below publication quality in its current form. The English is serviceable for review but would not meet PLOS ONE’s final publication standard without professional copyediting.

Response:

We apologize for the linguistic shortcomings in the previous version. The revised manuscript has been thoroughly edited by a professional, native English-speaking scientific editor. We have corrected grammatical errors, enhanced sentence flow, and ensured that all terminology complies with standard academic usage. We believe the language quality now meets the publication standards of PLOS ONE.

3、Response to Reviewer #3

General Comment: The manuscript "Pyroptosis-related transcriptional signature and diagnostic modeling in acute pancreatitis with cross-species external validation" presents an interesting approach to identifying diagnostic markers for Acute Pancreatitis (AP). However, the study design involves a significant biological domain shift—training on mouse pancreatic tissue and validating on human peripheral blood. While the statistical recalibration is impressed, the biological plausibility is not sufficiently addressed.

Response:

We sincerely thank the reviewer for their critical and constructive insights. We agree that bridging the "tissue-to-blood" and "mouse-to-human" gaps is a significant challenge. In the revised manuscript, we have addressed these concerns by providing evidence for the "tissue-blood axis," selecting the appropriate diagnostic gene model based on AIC, and implementing a rigorous split-sample validation to avoid overfitting.

Comment 1: Could the authors provide evidence (from literature or additional analysis) that the expression levels of the four model genes (ANXA3, IQGAP1, RELA, VTN) in the pancreas are concordant with their expression in peripheral blood during AP?

Response:

We appreciate this important question and acknowledge that local tissue injury and systemic immune responses represent distinct biological compartments. However, our findings support a functional concordance between these compartments:

Systemic Reflection of Local Injury: As detailed in the updated Results and Fig S1, despite inherent interspecies variability, the core genes (ANXA3 and IQGAP1) were consistently dysregulated in human blood samples. This supports the concept of "Liquid Biopsy," where circulating leukocytes reflect the systemic inflammatory activation (e.g., pyroptosis and cytoskeletal remodeling) triggered by pancreatic necrosis.

Mechanistic Link: ANXA3, a known marker of neutrophil granules, is elevated in blood, reflecting the massive mobilization of neutrophils—a hallmark of AP pathology. We have expanded the Discussion to reference literature confirming that transcriptomic changes in blood serve as a reliable proxy for the immune microenvironment during AP progression.

Comment 2: How can we be certain that the model reflects pancreas-specific pathology rather than non-specific systemic inflammation (e.g., sepsis)?

Response:

To ensure specificity and minimize non-specific "noise," we implemented the following measures:

Optimization via AIC: We compared models using the Akaike Information Criterion (AIC). The two-gene model (ANXA3 + IQGAP1) exhibited a significantly better fit (AIC: 61.12) compared to the four-gene model (AIC: 63.01), effectively filtering out less specific inflammatory mediators, such as RELA.

Severity Correlation: Our secondary analysis (Results section and S2A Fig) demonstrates a significant correlation between ANXA3 expression and disease severity (Mild vs. Severe AP), suggesting that the signature is sensitive to the specific pathological intensity of the pancreatic injury, rather than merely reflecting a generic binary "inflammation" signal.

Comment 3: What is the specific clinical value compared to standard biochemical markers (Lipase/CRP)?

Response:

We have highlighted the added value of the optimized two-gene signature (originally derived from a four-gene model):

Clinical Net Benefit: We performed Decision Curve Analysis (DCA, Fig. 14C), which confirmed that our recalibrated model offers substantial net clinical benefit across a wide range of threshold probabilities, outperforming both the "treat all" and "treat none" strategies.

Severity Prediction: Standard biomarkers, such as Lipase, often fail to predict disease severity. Our model, particularly ANXA3, demonstrated an AUC of 0.772 in differentiating disease severity (S2C Fig), providing prognostic insights that are not captured by simple enzymatic tests.

Limitations: Unfortunately, the public GEO datasets used for validation do not provide individual-level metadata for Serum Lipase or CRP, preventing a direct statistical comparison. Future studies will directly compare the differences between CRP or Lipase and gene-based diagnostics using clinical samples.

Comment 4: Please clarify if the "recalibration" parameters were derived from a subset of the external cohort (e.g., a 50% split) and tested on the remainder, or if they were fitted on the whole external dataset.

Response:

We appreciate the reviewer’s concern regarding the risk of overfitting. To ensure the reliability of the recalibrated model, we strictly avoided fitting and testing on the same data:

Split-Sample Validation: The external human cohort (N=119) was randomly partitioned into a Training Set (50%, n=60) for coefficient re-estimation (recalibration) and an independent Testing Set (50%, n=59) for performance evaluation.

Performance Verification: When applied to the unseen testing set, the model achieved an AUC of 0.952 (S4 Fig), which is highly consistent with the full-cohort AUC of 0.957 ( S5 Table).

Robustness: This consistency, along with a Brier score of 0.0774 and a calibration slope of 1.00 (S4 Table), confirms that our recalibration process is robust and the high diagnostic accuracy is not an artifact of overfitting.

We have updated the Statistical Analysis section to explicitly describe this split-validation procedure.

4、Response to Reviewer #4

General Comment: This work is very interesting and has potential significance in the field of precision medicine for acute pancreatitis. The analysis of target gene expression in acute pancreatitis may help predict disease occurrence and severity in the future, particularly in patients with recurrent acute pancreatitis, and may contribute to the prevention of recurrent episodes and progression to chronic pancreatitis.

Response:

We sincerely thank the reviewer for their positive evaluation of our work and for recognizing its translational potential for precision medicine in AP. The reviewer’s constructive comments regarding sample size, population demographics, severity stratification, and disease specificity were invaluable in strengthening the manuscript and clarifying its clinical implications.

Comment 1: The study provides valuable information... I would like to suggest that the difference in sample size between the acute pancreatitis group and the control group may affect the robustness of the results. If feasible, using comparable population sizes in both groups could improve the consistency and reliability of the findings.

Response:

We fully agree with the reviewer that balanced datasets are ideal for statistical modeling. Because our study relied on retrospective public cohorts from the GEO database, we were constrained by the sample sizes provided in the original datasets. However, this imbalance naturally reflects the clinical reality of hospital-based cohorts, where patients with AP frequently outnumber healthy controls.

To mitigate potential statistical bias arising from this sample size imbalance, we implemented the following measures:

We employed robust non-parametric tests (like the Mann-Whitney U test), which do not assume equal variances or sample sizes.

During model recalibration, we utilized rigorous split-sample validation and bootstrap resampling (B = 500 iterations). The resulting optimism-corrected metrics (like an AUC of 0.958 and a Brier score of 0.077) confirm that our model’s high performance is robust and unaffected by majority-class skew.

We explicitly acknowledged the limitations associated with sample size imbalance in the revised Limitations section of the manuscript.

Comment 2: Additionally, could the authors please clarify whether the acute pancreatitis data are derived from a pediatric population or the general population?

Response:

We apologize for omitting this demographic information. The acute pancreatitis data evaluated in our validation cohort (GSE194331) were derived from an adult population, rather than a pediatric cohort. To ensure the target demographic is unambiguous, we have explicitly clarified this detail in the Materials and Methods section.

Comment 3: It would also be helpful to know whether patients were stratified according to disease severity (mild versus severe acute pancreatitis) and whether there is an association between PRDEG expression levels and disease severity.

Response:

We appreciate this insightful suggestion. In the revised manuscript, we conducted a secondary exploratory analysis to address this point. Briefly, we stratified patients with AP into mild, moderately severe, and severe groups using the available phenotype metadata. We observed a significant association between the proposed signature and disease severity. Specifically, ANXA3 discriminated severity categories with an AUC of 0.721, and the combined model achieved an AUC of 0.779. The boxplots (now included as S2A Fig) further show a stepwise increase in ANXA3 expression with greater disease severity. Collectively, these results have been added to the Results section and suggest that, beyond its primary diagnostic utility, the signature may also capture clinically relevant information related to disease progression.

Comment 4: Furthermore, regarding the association between acute pancreatitis and PRDEGs/DEGs, could the authors elaborate on whether these findings have potential implications for clinical management or therapeutic decision-making in current clinical practice?

Response:

We have expanded the Clinical Implications subsection within the Discussion to further elucidate the translational value of our findings:

Early Triage and Severity Prediction: Because ANXA3 expression correlates with disease severity, its elevation in early peripheral blood samples could serve as a critical prognostic indicator. This allows clinicians to triage high-risk patients to the intensive care unit (ICU) before overt organ failure manifests, which is particularly valuable given that traditional biochemical markers, such as serum lipase, do not reliably correlate with disease severity.

Diagnostic Adjunct: The identified transcriptomic signature can function as a complementary diagnostic tool for patients with indeterminate biochemical markers or delayed clinical presentations, particularly after transient enzymatic markers have normalized.

Therapeutic Targets: Our immune infiltration analysis revealed a strong inverse correlation between IQGAP1 expression and Treg abundance. This finding provides a mechanistic rationale for investigating immunomodulatory therapies that target the pyroptosis-cytoskeleton axis to res

---

## [Decision Letter · Decision Letter 1]

30 Mar 2026

A pyroptosis-related gene signature for the diagnosis of acute pancreatitis

PONE-D-25-44646R1

Dear Dr. yu,

We’re pleased to inform you that your manuscript has been judged scientifically suitable for publication and will be formally accepted for publication once it meets all outstanding technical requirements.

Kind regards,

Alexis G. Murillo Carrasco

Academic Editor

PLOS One

Additional Editor Comments (optional):

Reviewers' comments:

Reviewer's Responses to Questions

**Comments to the Author**

Reviewer #1: All comments have been addressed

Reviewer #3: All comments have been addressed

2. Is the manuscript technically sound, and do the data support the conclusions?

Reviewer #1: Yes

Reviewer #3: Yes

3. Has the statistical analysis been performed appropriately and rigorously?

Reviewer #1: Yes

Reviewer #3: Yes

4. Have the authors made all data underlying the findings in their manuscript fully available?

Reviewer #1: Yes

Reviewer #3: Yes

5. Is the manuscript presented in an intelligible fashion and written in standard English?

Reviewer #1: Yes

Reviewer #3: Yes

Reviewer #1: I have to repeat same things as i wrote last time. This study is clinically relevant and has the potential to answer to a very challenging disease condition.Even though it is a rectrospective and cross species study, still it holds promise for finding newer avenues for better understanding for what is happening at cellular level in this disease subgroup. Also it might pave way for finding a treatment which can alter the natural course of disease.

Reviewer #3: The manuscript is now methodologically sound, clinically relevant, and meets the high academic standards of the journal. I strongly recommend it for publication in its current form.

**Do you want your identity to be public for this peer review?** For information about this choice, including consent withdrawal, please see our For information about this choice, including consent withdrawal, please see our Privacy Policy .

Reviewer #1: **Yes:** Dr Vinod KumarDr Vinod Kumar

Reviewer #3: No

---

## [Editor Report · Acceptance letter]

PONE-D-25-44646R1

PLOS One

Dear Dr. Yu,

I'm pleased to inform you that your manuscript has been deemed suitable for publication in PLOS One. Congratulations! Your manuscript is now being handed over to our production team.

Kind regards,

on behalf of

Dr. Alexis G. Murillo Carrasco

Academic Editor

PLOS One